# Feature learning from non-Gaussian inputs:
# the case of Independent Component Analysis in high dimensions

**Fabiola Ricci** [1]  **Lorenzo Bardone** [1]  **Sebastian Goldt** [1]

## Abstract

Deep neural networks learn structured features from complex, non-Gaussian inputs, but the mechanisms behind this process remain poorly understood. Our work is motivated by the observation that the first-layer filters learnt by deep convolutional neural networks from natural images resemble those learnt by independent component analysis (ICA), a simple unsupervised method that seeks the most non-Gaussian projections of its inputs. This similarity suggests that ICA provides a simple, yet principled model for studying feature learning. Here, we leverage this connection to investigate the interplay between data structure and optimisation in feature learning for the most popular ICA algorithm, FastICA, and stochastic gradient descent (SGD), which is used to train deep networks. We rigorously establish that FastICA requires at least $n \gtrsim d^4$ samples to recover a single non-Gaussian direction from $d$-dimensional inputs on a simple synthetic data model. We show that vanilla online SGD outperforms FastICA, and prove that the optimal sample complexity $n \gtrsim d^2$ can be reached by smoothing the loss, albeit in a data-dependent way. We finally demonstrate the existence of a search phase for FastICA on ImageNet, and discuss how the strong non-Gaussianity of said images compensates for the poor sample complexity of FastICA.

## 1. Introduction

The practical success of deep neural networks is generally attributed to their ability to learn the most relevant input features directly from data (LeCun et al., 2015). A classic example for this feature learning are deep convolu-

[1]International School of Advanced Studies, Trieste, Italy. Correspondence to: Fabiola Ricci <fricci@sissa.it>, Lorenzo Bardone <lbardone@sissa.it>, Sebastian Goldt <sgoldt@sissa.it>.

*Proceedings of the 42st International Conference on Machine Learning*, Vancouver, Canada. PMLR 267, 2025. Copyright 2025 by the author(s).

tional neural networks (CNNs), which recover Gabor filters (Gabor, 1946) like the ones shown in Figure 1(a) in their first layer when trained on sets of natural images like ImageNet (Krizhevsky et al., 2012; Lindsey et al., 2019; Guth & Ménard, 2024). Since Gabor filters are localised in both space and frequency domains, they are effective for capturing edges and textures in natural images (Hyvärinen et al., 2009). How these filters emerge from training a deep network end-to-end on natural images with stochastic gradient descent remains a key question for the theory of neural networks.

Deep neural networks are not the only approach to learning Gabor filters directly from data. Instead, it is well-known that independent component analysis (ICA) (Comon, 1994; Bell & Sejnowski, 1995; Olshausen & Field, 1996; Bell & Sejnowski, 1996; Hyvärinen & Oja, 2000), a simple, unsupervised learning algorithm, learns filters that are similar to the first-layer filters of a deep CNN when applied to patches of natural images, see Figure 1(b). The key idea of ICA is to identify directions of maximal non-Gaussianity in input space, suggesting that the non-Gaussian structure in data plays a crucial role in shaping the representations learnt by deep neural networks (Refinetti et al., 2023). More precisely, given a set of $n$ zero-mean inputs $\mathcal{D} = \{x^1, x^2, \ldots, x^n\}$, ICA seeks the direction $w^* \in \mathbb{R}^d$ that maximises the non-Gaussianity of the projections of the inputs $s := w \cdot x$,

$$w^* := \arg\max_{\|w\|=1} \mathbb{E}_{\mathcal{D}} \, G(w \cdot x), \tag{1}$$

where the contrast function $G$ is a measure of the non-Gaussianity of the projection $s$, for example its excess kurtosis $G(s) = s^4 - 3$. For ICA, inputs are always pre-whitened, so ICA can be seen as a refinement of standard principal component analysis, which instead seeks the projection of the data that maximises its variance, i.e. $G(s) = s^2$. On images, PCA yields a completely different set of filters which is spatially extended and oscillating due to the approximate translation-invariance of the image patches (Hyvärinen et al., 2009), see Figure 1(c).

A closer look at the dynamics of the CNNs trained on the standard ImageNet dataset (Deng et al., 2009) reveals further similarities between ICA and deep CNNs. In Figure 1(d), we plot the excess kurtosis of the first-layer pro-

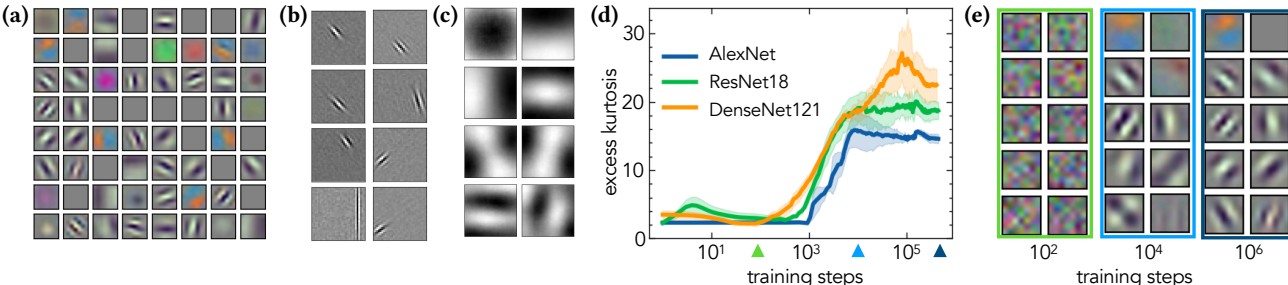

*Figure 1.* **Independent component analysis yields similar filters as deep convolutional neural networks. (a)** The first-layer convolutional filters extracted from an AlexNet trained on ImageNet show the hallmarks of Gabor filters (Gabor, 1946), like orientation, localisation, and bandwidth-specificity. **(b)** Independent components found with online SGD on 100 000 $64 \times 64$ patches extracted from ImageNet. **(c)** Leading principal components of the same ImageNet patches. **(d)** Excess kurtosis of the representations of the first convolutional layer of three deep CNNs during training on ImageNet. For each filter, we compute the dot product $s$ between its weights and patches sampled from ImageNet, then compute its excess kurtosis averaged over image patches and neurons. **(e)** Snapshots of a subset of convolutional filters of AlexNet at various points during training. As the excess kurtosis of representations increases, prototypes of Gabor filters emerge, which are then sharpened during the remainder of training. Full experimental details in Appendix A.1.

jections $s_\mu^k = w^k \cdot x^\mu$ of ImageNet patches $x^\mu$ along the weights $\{w^k\}_{k=1,\dots,K}$ of the $K$ first-layer filters of three different CNNs during training, averaged over patches and neurons. If the projections $s_\mu^k$ are normally distributed, the excess kurtosis is (close to) zero, as is the case early during training when weights are close to their (uniform) initialisation. In all three networks, the non-Gaussianity of projections increases sharply between $10^3$ and $10^4$ steps, before plateauing. Interestingly, the moment in which the deep CNN focuses on non-Gaussian projections is precisely the moment in which the Gabor filters form, see Figure 1(e).

Taken together, the similarity between filters obtained by deep CNNs and ICA and the dynamics of the excess kurtosis of first-layer representations of deep CNNs suggest that ICA can serve as a simplified, yet principled model for studying feature learning from non-Gaussian inputs.

In this work, we develop a quantitative theory of feature learning from non-Gaussian inputs by providing a sharp analysis of the sample complexity of two algorithms for Independent Component Analysis, namely FastICA, the most popular ICA algorithm used in practice, and SGD, which is used to train deep neural networks.

Early theoretical works on independent component analysis focused mostly on the speed of convergence of different algorithms for finding $w^*$ in the classical regime of fixed input dimension and large number of samples; the most popular algorithm for performing ICA, FastICA (Hyvärinen & Oja, 2000), derives its name from the fact that it converges with a quadratic (Hyvarinen, 1999) rather than linear rate. However, ICA struggles in modern applications where inputs tend to be high-dimensional; for example, its performance is highly sensitive to its initial conditions (Zarzoso et al., 2006; Auddy & Yuan, 2024). For algorithms running in high dimensions, the main bottleneck is typically

not the speed of convergence, but the length of the initial search phase before the algorithm recovers any trace of a signal (Bottou, 2003; Bottou & Le Cun, 2003). The focus of our work is therefore in establishing lower bounds on the sample complexity required by FastICA and SGD for escaping the search phase.

Recently, Auddy & Yuan (2024) showed that in high dimensions, $n \gtrsim d/\varepsilon^2$ samples are necessary and sufficient to ensure the existence of estimates of the non-Gaussian directions with an error up to $\varepsilon$ when given unlimited computational resources, while the sample complexity required for polynomial time algorithms is $n \gtrsim d^2/\varepsilon^2$. They also proposed an initialisation scheme for FastICA that reaches the optimal performance. While these results establish fundamental statistical and computational limits of ICA, a precise and rigorous analysis of how the two main algorithms used in practice, FastICA and SGD, escape the search phase starting from random initialization is lacking.

In this paper, we derive sharp algorithmic thresholds on the sample complexity required by ICA to recover non-Gaussian features of high-dimensional inputs. Our analysis leverages recent breakthroughs in the analysis of supervised learning dynamics with Gaussian inputs Ben Arous et al. (2021); Damian et al. (2023); Dandi et al. (2024) for analysing the unsupervised case with non-Gaussian inputs. Our main results are as follows:

- We prove that the popular FastICA algorithm exhibits poor sample complexity, requiring a large-size batch of $n \gtrsim d^4$ to recover a hidden direction in the first step (see Section 3.3);

- We prove that the sample complexity of online SGD can be reduced down to the computational threshold of $n \gtrsim d^2$, at the cost of fine-tuning the contrast function in a data-dependent way (see Section 3.5);

- We demonstrate that FastICA exhibits a search phase at linear sample complexity when trained on ImageNet patches, but recovers a non-Gaussian direction at quadratic sample complexity, and we discuss how the strong non-Gaussianity of real images speeds up recovery (see Section 4).

The main technical challenge in establishing our results is dealing with the non-Gaussianity of the inputs, which adds complexity to our analysis both by introducing additional terms whose statistics need to be controlled, and by impacting the intrinsic properties of the loss function.

## 2. Setup

We first describe the most popular algorithm to perform ICA, called FastICA, and we introduce the classic ICA model for synthetic data which will allow us to systematically test the performance of FastICA.

### 2.1. FastICA and contrast functions

Performing ICA on a generic set of inputs $x \in \mathbb{R}^d$ drawn from a data distribution $\mathbb{P}$ with zero mean and identity covariance means finding a unit vector $w \in \mathbb{S}^{d-1}$ which is an extremum of the *population* loss

$$\mathcal{L}(w) := \mathbb{E}_{\mathbb{P}} \, G(w \cdot x), \tag{2}$$

where $G : \mathbb{R} \to \mathbb{R}$ is a suitable "contrast function" which measures the non-Gaussianity of the projections $s = w \cdot x$. The two standard choices for $G(s)$ used in practice (Pedregosa et al., 2011) are

$$G(s) := -e^{-s^2/2} \quad \text{and} \quad G(s) := \frac{\log \cosh(as)}{a} \tag{3}$$

for some $a \in [1, 2]$. Another classical choice is the excess kurtosis, $G(s) := s^4 - 3$, even though it is more sensitive to outliers; therefore, the first two contrast functions are preferred in practice since their growth is slower than polynomial. In practice, the expectation is approximated as an average over a set of $n$ centered and whitened inputs $\mathcal{D} = \{x^\mu\}_{\mu=1}^n$ (that we assume to be drawn i.i.d. from $\mathbb{P}$).

As an optimisation algorithm, we focus on the classic FastICA algorithm of Hyvärinen & Oja (2000). While a number of works have recently proposed alternative algorithms with provable guarantees, e.g. Arora et al. (2012), Anandkumar et al. (2014), or Voss et al. (2015), we focus instead on FastICA since it is the most popular ICA algorithm (it is the reference implementation in scikit-learn (Pedregosa et al., 2011)).

FastICA finds extremal points of the population loss (2) via a second-order fixed-point iteration. Here, we simply state the FastICA algorithm for convenience; see Hyvärinen & Oja (2000) for a detailed derivation and discussion. Given a dataset $\mathcal{D}$, we initialise the weight vector randomly on the unit sphere, $w_0 \sim \text{Unif}(\mathbb{S}^{d-1})$, and then iterate the **FastICA updates** for $t \geq 1$ until convergence:

$$\begin{cases} \widetilde{w}_t & = \mathbb{E}_{\mathcal{D}}[x \, G'(w_{t-1} \cdot x)] - \mathbb{E}_{\mathcal{D}}[G''(w_{t-1} \cdot x)]w_{t-1}, \\ w_t & = \widetilde{w}_t / \|\widetilde{w}_t\|. \end{cases} \tag{4}$$

The FastICA iteration is made of two contributions: a "gradient" term $\nabla_w \mathcal{L}(w, x)\big|_{w=w_{t-1}} = x \, G'(w_{t-1} \cdot x)$, which drives the algorithm towards a direction where the gradient vanishes, and the "regularisation" $\mathbb{E}_{\mathcal{D}}[G''(w_{t-1} \cdot x)]w_{t-1}$, which ensures quadratic rather than the linear convergence of first-order methods, giving FastICA its name. As we will show in Proposition 1, the regularisation term is also key for efficient learning.

### 2.2. The ICA data model

To carefully test the sample complexity of FastICA and other ICA algorithms like SGD in a controlled setting, we first study the case of synthetic data according to a noisy version of the standard ICA data model before moving to real data in Section 4. The idea is to have inputs that follow an isotropic Gaussian distribution in all but one direction that we will call the *spike* $v \in \mathbb{S}^{d-1}$. The projection of the inputs along the spike yields a non-Gaussian distribution, so this is the direction that ICA should recover. Specifically, we assume that the data points $\mathcal{D} = \{x^\mu\}_{\mu=1}^n$ are drawn according to

$$x^\mu = S\left(\sqrt{\beta}\, \nu^\mu v + z^\mu\right) \in \mathbb{R}^d, \tag{5}$$

where $z^\mu \sim \mathcal{N}(0, \mathbb{1}_d)$ is a vector with white noise, while the scalar random variable $\nu^\mu$ is the latent variable for each input, and follows a non-Gaussian distribution. For concreteness, we choose a Rademacher distribution under which $\nu^\mu = \pm 1$ with equal probability. The signal-to-noise ratio $\beta \geq 0$ sets the relative strength of the non-Gaussian component along $v$ compared to the Gaussian part $z^\mu$ of each input. Since whitening the inputs is a standard preprocessing step for FastICA (Hyvärinen & Oja, 2000), we finally pre-multiply the inputs with the whitening matrix

$$S = \mathbb{1}_d - \frac{\beta}{1 + \beta + \sqrt{1 + \beta}} vv^\top \in \mathbb{R}^{d \times d},$$

which ensures that the covariance of the inputs is simply the identity. This whitening step makes it impossible to trivially detect the non-Gaussian direction $v$ by performing PCA or any other method which is based only on the information in the first and second cumulant. An alternative perspective of the ICA model is that it provides inputs where the noise is correlated, and in particular has a negative correlation with the spike to be recovered.

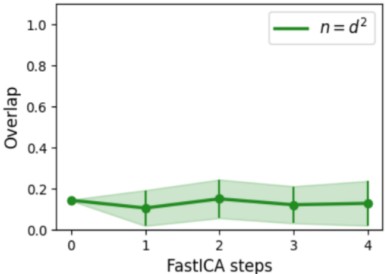 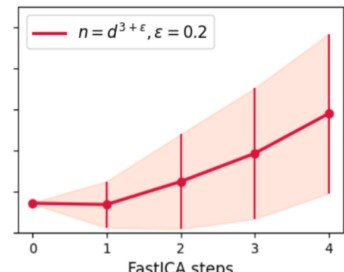 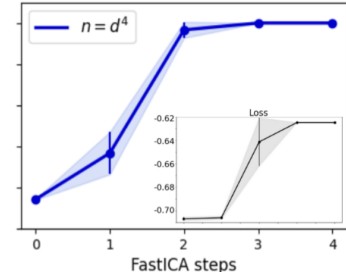

*Figure 2.* **Performance of FastICA on the spiked cumulant model for various sample complexity regimes.** We run the standard FastICA updates (4) on data drawn from a noisy ICA model (5). At each step of the algorithm, we draw a new batch of size $n = d^2$ (**left**), $n = d^{3+\varepsilon}$ with $\varepsilon = 0.2$ (**middle**) and $n = d^4$ (**right**). In the latter case, we also show the corresponding loss function. We plot the overlap between the planted non-Gaussian direction $v$ and the estimate $w$ produced by FastICA. While recovery of the spike is theoretically possible at $n = d^2$ (see Section 2.3), FastICA does not recover the spike at that sample complexity, and instead requires $n = d^4$ samples to recover the spike $v$ reliably. Theorem 2 confirms this picture for a general data model. Full experimental details in Appendix A.2 and additional cases $\varepsilon = 0.5, 0.8$ in Figure 5.

## 2.3. Fundamental limits of recovery

We now briefly recall the fundamental limits of recovering the spike in the ICA data model (5), which will serve as benchmarks to evaluate the performance of specific algorithms like FastICA or SGD. From an information-theoretic point of view, i.e. when given unbounded computational resources, the presence of the non-Gaussian direction $v$ can already be detected using a linear number of samples $n \gtrsim d$. However, detecting the spike at this sample complexity is only possible through exponentially complex algorithms that are based on exhaustive search. Efficient algorithms, that can run in polynomial time in $d$, require at least a quadratic number of samples $n = \omega(d^2)$.

For details on the algorithmic result, see Szekely et al. (2024); Dudeja & Hsu (2024) and Auddy & Yuan (2024) for a thorough analysis using low-degree methods, and Diakonikolas et al. (2017) for a Statistical Query (SQ) analysis, which reaches equivalent predictions using the low-degree to SQ equivalence (Brennan et al., 2021). Auddy & Yuan (2024) establishes the information-theoretic limits of reconstructing the spike using a min-max type analysis. A similar statistical-to computational gap appears in the related problem of tensor PCA (Richard & Montanari, 2014).

## 2.4. Experiment: FastICA requires a lot of data to recover the non-Gaussian direction

We are now in a position to evaluate the FastICA algorithm in the controlled setting of the noisy ICA model (5), where the goal is to recover the planted non-Gaussian direction (or "feature") $v$. Specifically, we can compare the sample complexity of FastICA, i.e. the number of samples FastICA requires to recover $v$, to the fundamental limits of efficient or polynomial-time recovery, which is $n \gtrsim d^2$.

In Figure 2, we plot the absolute value of the overlap between the spike $v$ and the estimate $w$ obtained by running

FastICA for a couple of steps with $n = d^\theta$ samples. In the plot, we consider the limit of online learning, in the sense that we draw a new training set at each step of the algorithm; in Figure 7, we show that the picture does not change when using the same training set repeatedly; see also our discussion in Section 3.3.3.

We can see immediately that at quadratic sample complexity, FastICA does not recover the spike at all. Instead, the overlap is stuck at a value of roughly $1/\sqrt{d}$, which is the typical overlap of a randomly direction in $\mathbb{R}^d$ with the spike. This small value for the overlap suggests that the algorithm is stuck in the search phase, and does not recover the spike at quadratic sample complexity. With $n = d^4$ samples instead, FastICA recovers the spike within a few steps. In the intermediate regime of $n = d^{3+\varepsilon}$ samples, $\varepsilon = 0.2$, the algorithm recovers the spike more slowly, and there are large fluctuations.

These experiments suggest that FastICA has a poor sample complexity that is quartic in the input dimension for recovering the spike, with an intermediate regime at $n = d^{3+\varepsilon}$ with $\varepsilon \in (0, 1)$. Our theoretical analysis of FastICA will now confirm that these are indeed the fundamental limits of FastICA.

## 3. Theoretical Results

Before diving into our analysis of FastICA and stochastic gradient descent (SGD) for performing ICA, we highlight how to address the key technical difficulty in analysing learning from non-Gaussian inputs.

### 3.1. ICA, likelihood ratios, and information exponents

The key difficulty in analysing the ICA loss (2) or the FastICA update (4) is that it entails an average over an explicitly non-Gaussian distribution of the inputs. This is in

contrast to the vast majority of works on (supervised) learning algorithms, which usually consider a Gaussian input distribution, an input distribution that is effectively Gaussian via the Gaussian Equivalence Theorem, or inputs that are distributed uniformly on the unit hypercube (as discussed at the end of the section).

The key idea of the theoretical analysis of FastICA and SGD is to rewrite the average over the non-Gaussian inputs as an average over the standard normal distribution $\mathbb{P}_0 = \mathcal{N}(0, \mathbb{1}_d)$, as

$$\mathcal{L}(w) := \mathbb{E}_{\mathbb{P}}[G(w \cdot x)] = \mathbb{E}_{\mathbb{P}_0}[G(w \cdot x)\ell(v \cdot x)] \quad (6)$$

by introducing the *likelihood ratio*

$$\ell(s) := \frac{d\mathbb{P}}{d\mathbb{P}_0}(s), \quad (7)$$

akin to how one would proceed in an analysis with the second moment method in hypothesis testing (Kunisky et al., 2019). Intuitively, the likelihood ratio (or more precisely, its norm) measures how different the distribution of the projection $s = w \cdot x$ is from a standard Gaussian distribution. The likelihood ratio will thus be the key object in our study to determine the number of samples that a given algorithm like FastICA requires to find a projection that is distinctly non-Gaussian and yields a large likelihood ratio.

Under the noisy ICA model (5), the likelihood ratio (7) depends only on the projection of the inputs along the non-Gaussian direction $v$ to be learnt. We can expand the population loss in terms of a single scalar variable, the overlap $\alpha := w \cdot v$ between weight $w$ and spike $v$:

$$\mathcal{L}(w) = \mathbb{E}_{\mathbb{P}_0}[G(w \cdot x)\ell(v \cdot x)] = \sum_{k \geq 0}^{\infty} \frac{c_k^G c_k^\ell}{k!}\alpha^k, \quad (8)$$

where the last equality has been obtained by expanding the contrast function $G$ and the likelihood ratio $\ell$ in series of Hermite polynomials and by using the orthogonality of Hermite polynomials (Lemma 11 in Appendix B.3). A key quantity for the analysis of ICA is the order $k^*$ of the first term in the expansion (8) that has a non-zero coefficient. This quantity was introduced as the "*information exponent*" by Ben Arous et al. (2021) in the context of supervised learning, when they showed that it governs the sample complexity of online SGD in single-index models, like the noisy ICA model. Related quantities like the *leap index* of Abbé et al. (2023); Dandi et al. (2024) and the *generative exponent* of Damian et al. (2024) govern the sample complexity for more complex data models with several directions to be learnt. Here, we note that due to the whitening of the inputs, the population loss (2) for the noisy ICA model model has information exponent $k^* = 4$, as we show in Appendix C.1.

### 3.2. Main assumptions

For the following theoretical analysis, we assume that:

**Assumption 1.** *The contrast function $G$ and the data distribution $\mathbb{P}$ are such that the loss* (2) *is a function of the overlap $\alpha := w \cdot v$, for a given spike $v$ with unit norm. Moreover, we assume that $\ell(\alpha) = \frac{d\mathbb{P}}{d\mathbb{P}_0}(\alpha) \in \mathrm{L}^2(\mathbb{R}, \mathbb{P}_0)$.*

This assumption is trivially satisfied for the noisy ICA model (5) and the standard contrast functions in Equation (3), but it allows us to prove our results on the sample complexity of FastICA (Theorem 2) and smoothed SGD (Theorem 4) for arbitrary sub-Gaussian latent variables $\nu$ (see Definition 6 in Appendix B). In principle, these hypotheses exclude the case of a fat-tailed distributions on the latent variable, i.e. those that do not belong to $\mathrm{L}^2(\mathbb{R}, \mathbb{P}_0)$. However, the results of simulations with a Laplace prior on $\nu$ (see Figure 9) reveal the same picture as in the sub-Gaussian case, suggesting that this assumption could be further relaxed.

**Assumption 2** (Contrast function). *We assume $G$ to be even. For our analysis of FastICA, we require $G \in C^3(\mathbb{R})$ with bounded derivatives. For smoothed SGD in Section 3.5, we additionally require $\sum_{k \geq 0}^{\infty}(c_k^G)^2/k! < 1$ and that there exist some $C_1, C_2 > 0$ such that, for any $s \in \mathbb{R}$, $|G'(s)| \leq C_1(1 + s^2)^{C_2}$ holds.*

### 3.3. Sample complexity of FastICA

Our goal is to rigorously quantify the sample complexity of FastICA on the noisy ICA model (5). FastICA proceeds by performing a few iterations of the update (4) with the full data set. This setting is precisely the large-batch (Ba et al., 2022; Damian et al., 2022) or "giant step" (Dandi et al., 2024) regime that has attracted a lot of theoretical interest recently. Following the approach of these recent works, the key idea of our analysis is to study whether a single step of FastICA with a large number of samples $n \asymp d^\theta$ yields an estimate $w_1$ that has an overlap with the spike $v$ that does not vanish with the input dimension.

#### 3.3.1. THE IMPORTANCE OF REGULARISATION

It is instructive to first consider briefly the ideal case of infinite data. The following Proposition 1 shows that FastICA is able to fully recover the spike $v$ as $d \to \infty$ if we replace the empirical average $\mathbb{E}_\mathcal{D}$ with the population average $\mathbb{E}_\mathbb{P}$. More interestingly, we show that it is the regularisation term, originally introduced to speed up convergence (see Section 2.1), that gives the essential contribution to escape mediocrity at initialisation.

**Proposition 1** (Infinite batch size). *Assume that the inputs are distributed according to the noisy ICA model* (5) *and that $c_4^G \neq 0$ - this holds for the standard contrast functions in Equation* (3). *Set $\alpha_1 := w_1 \cdot v$, where $w_1$ is the updated weight vector given by the first iteration of FastICA.*

*Then, in the infinite batch-size limit, we have that*

$$\alpha_1^2 = 1 - o(1).$$

*On the other hand, if the regularisation term is missing and only the "gradient step" is taken, i.e. the iteration reads $\widetilde{w}_t = \mathbb{E}[x\, G'(w_{t-1} \cdot x)]$, then we obtain $\alpha_1^2 = O(1/d)$.*

### 3.3.2. FINITE SAMPLES: HARDNESS OF LEARNING

We now provide almost sharp bounds on the sample complexity of FastICA for learning the non-Gaussian feature $v$. For the analysis, we follow the recent "one-step" analyses (Ba et al., 2022; Damian et al., 2022; Dandi et al., 2024) and study the overlap of the spike with the weight obtained from performing a single step of FastICA with a large batch of $n = d^\theta$ samples. More precisely, we provide a lower bound (positive results) for the amount of signal that can be weakly recovered given a number of samples on the order of $d^{k^*}$, where $k^*$ is the information exponent of the loss, and precisely $k^* = 4$ for the noisy ICA model. In the data-scarce regime where $n = o(d^{k^*})$, we provide upper bounds (negative results) on the overlap $\alpha = w \cdot v$ that can be achieved in a single step of FastICA. We distinguish two cases: either there is no improvement with respect to the random initialisation, or the upper bound on the amount of signal learned is dimensionality-dependent, meaning that $n = \Theta(d^{k^*-1})$ are still not sufficient to exit from the search phase in the high-dimensional limit. Yet this regime smoothly bridges the situation of total ignorance, where nothing is learnt at all, and recovery of the spike in one step, thanks to a continuous dependence on the parameter $\delta$.

**Theorem 2.** *(Finite batch size) Let $k^*$ be the information exponent of the population loss* (2). *Consider a number of samples $n = \Theta(d^{k^*-\delta})$, for $\delta \in [0,2]$. Set $\alpha_1 := w_1 \cdot v$, where $w_1$ is the updated weight vector given by the first iteration of FastICA. Then,*

$$\begin{cases} \delta \in (1,2] & \Rightarrow \quad \alpha_1^2 = O\left(\frac{1}{d}\right), \\ \delta \in (0,1] & \Rightarrow \quad \alpha_1^2 = O\left(\frac{1}{d^\delta}\right), \\ \delta = 0 & \Rightarrow \quad \alpha_1^2 \geq 1 - o(1). \end{cases}$$

We prove Theorem 2 in Appendix D. Even if Theorem 2 is not specific to the noisy ICA model, and instead applies to any input model and contrast function satisfying Assumption 1 and 2, it is instructive to look at the special case of Rademacher prior on the latent variable and standard contrast function, Equation (3). In this setting, we have proved in Appendix C.1 in the appendix that $k^* = 4$. Therefore, we find that $n = \Theta(d^4)$ is sufficient to perfectly recover the spike with a single step, up to contributions which vanish when $d$ goes to infinity, in line with what we see in the experiments of Figure 2. Moreover, if $d^3 \lesssim n \ll d^4$, it turns out that the overlap $\alpha_1 = w_1 \cdot v$ is bounded by a dimension-dependent constant, which explains the slow increase of the

overlap in that regime. For quadratic sample complexity, the gradient in the FastICA update does not concentrate, so the algorithm does not recover the signal and remains stuck in the search phase.

*Key elements of the proof.* The overall ideas of the proof are similar to those employed in the analysis of the teacher-student setup of Theorems 1 and 2 from (Dandi et al., 2024), with the following main differences. First of all, in our unsupervised case, instead of isotropic inputs, the data points are in general non-Gaussian distributed; as already observed in Section 2.3, this impacts on the information exponent $k^*$ of the population loss. Second, unlike for SGD, the iteration of FastICA presents both a gradient term and a regularisation term which is data-dependent (the regularisation "constant" depends on $G''(w \cdot x)$, and its analysis is hence just as complex as that of the gradient term). The general strategy is computing the expectations for both the signal $\widetilde{w}_1 \cdot v$ and the noise $\|\widetilde{w}_1\|$, and then identifying how much data $n$ allows for the concentration of the two quantities. To be able to compute the expectations, we exploit the orthogonality properties of Hermite polynomials proved in Corollary 14. Once concentration of both signal and noise is established, the positive result corresponds to the signal dominating the noise, and vice versa for the intermediate regime in the middle. Conversely, if the signal does not concentrate, $n$ is not sufficiently large for the overlap to escape from its scaling at initialisation and first upper bound holds. □

### 3.3.3. DISCUSSION

The almost sharp analysis of the high-dimensional asymptotics of FastICA in Theorem 2 is our first main result, and explains theoretically why FastICA tends to struggle in high dimensions (Zarzoso et al., 2006; Auddy & Yuan, 2024). While FastICA is often applied after reducing the dimension of the inputs using PCA, it should be noted that the scaling of $n \asymp d^4$ on the noisy ICA model is poor even in moderate to small dimensions. One non-standard way to escape this scaling is to use the ad-hoc initialization scheme proposed recently by Auddy & Yuan (2024), which allows to find an initialisation that gives ICA a warm start with a non-trivial overlap given only $n \gtrsim d^2$ samples. However, there remains a large gap between the performance of FastICA as it is commonly used, and the $n \gtrsim d^2$ or, more generally speaking, $n \gtrsim d^{k^*/2}$ bounds we have from low-degree methods, see (Damian et al., 2024). One may therefore ask whether other algorithms, and in particular the simple stochastic gradient descent, may be more efficient at performing ICA, not least since neural networks trained with stochastic gradient descent seem to learn ICA-like filters in their first layer rapidly.

## 3.4. Vanilla SGD

In order to take a closer look at feature extraction in the context of deep neural networks, we analyse SGD as an alternative algorithm to perform ICA. Consider a set of $n$ centered and whitened data points $\mathcal{D} = \{x^\mu\}_{\mu=1}^n$. We sample a new data point at each step. For a suitable learning rate $\delta > 0$, each iteration of the spherical online SGD, for $w_0 \sim \text{Unif}(\mathbb{S}^{d-1})$, is defined as

$$\begin{cases} \widetilde{w}_t = w_{t-1} + \frac{\delta}{d} \nabla_{\text{sph}} \mathcal{L}(w, x_t)\big|_{w=w_{t-1}} & t \geq 1, \\ w_t = \widetilde{w}_t / \|\widetilde{w}_t\|, \end{cases}$$

The spherical gradient $\nabla_{\text{sph}}$ for a function $f$ is given by $\nabla_{\text{sph}} f(w, \cdot) := (\mathbb{1}_d - ww^\top) \nabla_w f(w, \cdot)$. In the case of the noisy ICA model model (5), we can directly apply the analysis performed by Ben Arous et al. (2021), which guarantees that if $n = \Omega(d^3 \log^2 d)$, or more in general $n = \Omega(d^{k^*-1} \log^2 d)$, the spike is "strongly" recovered, meaning that $w_n \cdot v \to 1$ in probability for $n$ going to infinity. Moreover, if $n = O(d^3)$, the spike cannot be recovered, i.e. $\sup_{t \leq n} |v \cdot w_t| \to 0$ in probability.

Hence, vanilla online SGD is already faster that FastICA in the large-batch setting. However, given the information-theoretic results of Section 2.3, online SGD does not achieve the optimal performance and there remains a statistical-to-computational gap.

## 3.5. Closing the statistical-to-computational gap: smoothing the landscape

We now show that the statistical-to-computational gap of SGD on noisy ICA can be closed by running online spherical SGD on a *smoothed loss* following the approach of Biroli et al. (2020); Damian et al. (2023):

**Definition 3.** *(Biroli et al., 2020; Damian et al., 2023) For $\lambda \geq 0$ and $x \in \mathbb{R}^d$, the smoothed loss function is $L_\lambda(w, x) := \mathcal{L}_\lambda[G(w \cdot x)]$, where the smoothing operator $\mathcal{L}_\lambda$ reads*

$$\mathcal{L}_\lambda[G(w \cdot x)] := \mathbb{E}_{z \sim \mu_w}\left[G\left(\frac{w + \lambda z}{\|w + \lambda z\|} \cdot x\right)\right],$$

*if $\mu_w$ is the uniform distribution over $\mathbb{S}^{d-1}$ conditioned on being orthogonal to $w$.*

The key intuition behind the smoothing operator is that, thanks to large $\lambda$ (than scales with the input dimension), it allows to evaluate the loss function in regions that are far from the iterate weight $w_t$, collecting non-local signal that alleviates the flatness of the saddle around $\alpha = 0$ of $\mathcal{L}$, reducing the length of the search phase; we illustrate the effect of smoothing on the loss in Figure 6. For implementation purposes, we give an explicit formula for the smoothed gradient of said loss in Appendix E. Note that the "large steps" allowed by a large $\lambda$ help in escaping the search phase.

Biroli et al. (2020) introduced the idea of smoothing in the context of tensor PCA. Damian et al. (2023) showed that smoothing the loss for a single neuron $\sigma(w \cdot x)$ with activation function $\sigma : \mathbb{R} \to \mathbb{R}$ trained on a supervised task with Gaussian inputs and labels provided by a teacher neuron $\sigma(v \cdot x)$ reduces the sample complexity of weak recovery for online SGD from $d^{k^*-1}$ (Ben Arous et al., 2021) down to the optimal $d^{k^*/2}$. Their analysis crucially relies on the fact that teacher and student have the same activation function $\sigma$. For ICA, one can roughly think of the contrast function as the student activation function, which will in general have a different information exponent from the loss function, which depends also on the likelihood ratio. We therefore generalise the analysis of Damian et al. (2023) to this mismatched case and show that the dynamics of smoothed SGD on ICA is governed by the interplay of the information exponents of the population loss and the contrast function respectively, called $k_1^*$ and $k_2^*$, and that the optimal sample complexity is reached if and only if $k_1^* = k_2^*$. This fact follows immediately from the heuristic analysis performed in Section 3.5.1, where it is shown that for the optimal $\lambda = d^{1/4}$, the ODE describing the behavior of the overlap $\alpha$ at the beginning of learning is

$$\alpha'(t) = \frac{\alpha(t)}{d^{\boldsymbol{k_1^* - k_2^*}/2}}. \tag{9}$$

On the noisy ICA model (5), for which $k_1^* = 4$, we thus find the following picture. For the standard contrast functions (Equation (3)), $k_1^* \neq k_2^* = 2$, and then smoothed online SGD does not improve the sample complexity of vanilla online SGD: the spike is still recovered in a cubic time. In contrast, if $k_2^* = 4$, i.e. the information exponents of the loss and the contrast function are equal, smoothed online SGD is able to recover the spike in a quadratic time, which means that smoothing the landscape reduces the sample complexity of online SGD down to the optimal threshold suggested by low-degree methods. Even if it would be clearly necessary a fine-tuning of the contrast function in a data-dependent way which cannot be performed in practice, we observe that in our case the fourth-order Hermite polynomial $G(s) = h_4(s)$, which for whitening data is nothing but the excess kurtosis $G(s) = s^4 - 3$, is indeed an optimal contrast functions. Remarkably, the kurtosis is the most classical measure of non-Gaussianity that one can find in the literature (see e.g. Hyvärinen & Oja (2000)) and it has been employed to study recent developments of FastICA in (Auddy & Yuan, 2024).

### 3.5.1. HEURISTIC DERIVATION

We first heuristically derive the sample complexities of Theorem 4, following Damian et al. (2023). We will use Corollary 13 to deal with expectations of multiple products of Hermite polynomials appearing due to non-Gaussian inputs.

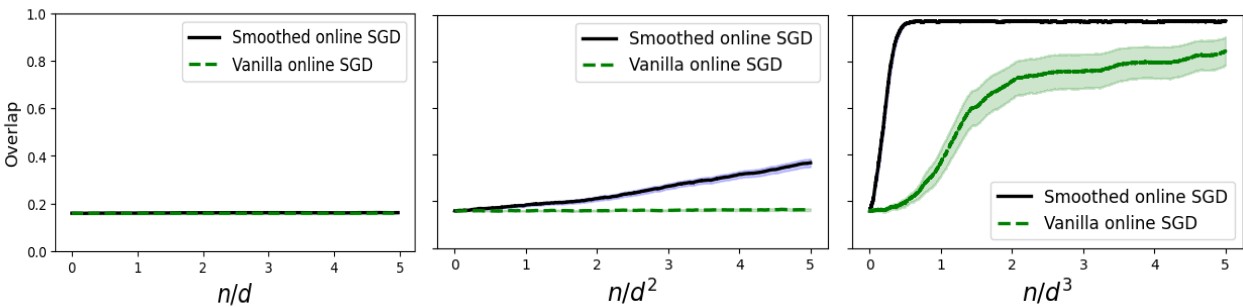

*Figure 3.* **Performances of smoothed online SGD and vanilla SGD on the noisy ICA model, for** $G(s) = h_4(s)$ **and** $G(s) = -\exp(-s^2/2)$ **respectively. Left:** In a linear time, nor smoothed and vanilla SGD recover the spike. **Middle:** In the quadratic regime, smoothed SGD recovers, although vanilla SGD does not. **Right:** In the cubic regime, both smoothed and vanilla SGD recover the spike. Experimental details in Appendix A.3.

A crucial quantity is the signal-to-noise SNR since, as shown in Damian et al. (2023), at the beginning of leaning the overlap $\alpha := w \cdot v$ is given by the ODE $\alpha'(t) = \text{SNR}/\alpha(t)$ with

$$\text{SNR} := \frac{\mathbb{E}[g \cdot v]^2}{\mathbb{E}[\|g\|^2]},$$

where $g$ is the online spherical gradient $g := \nabla_{\text{sph}} L_\lambda(x, w)$. We compute how the SNR scales with the dimension of the inputs $d$.

**Signal** Recall the scaling computed by Damian et al. (2023) for the signal in the case of an even contrast function:

$$\mathbb{E}[g \cdot v] = \Theta(\alpha \, d^{-(k_1^* - 2)/2}/\lambda^2),$$

when $\alpha \leq \lambda d^{-1/2}$. Note that it depends on the information exponent $k_1^*$ of the loss.

**Noise** We look at the scaling of the noise $\mathbb{E}[\|g\|^2]$. By definition, $g = \Theta\left(\lambda^{-1} x \, \mathcal{L}_\lambda(G'(w \cdot x))\right)$. Since with high probability $\|x\|^2 = O(d)$, we study the second moment of $s := \mathcal{L}_\lambda(G'(w \cdot x))$. By exploiting the definition of the smoothing operator and writing the expectation in terms of the likelihood ratio, we get that, for two replicas $z, z' \sim \mu_w$,

$$\mathbb{E}\left[s^2\right] = \mathbb{E}\left[\mathbb{E}_{z,z'}\left[G'\left(\frac{w + \lambda z}{\sqrt{1 + \lambda^2}} \cdot x\right) G'\left(\frac{w + \lambda z'}{\sqrt{1 + \lambda^2}} \cdot x\right)\right]\right]$$

$$= \mathbb{E}_{z,z'}\left(\mathbb{E}_{\mathbb{P}_0}\left[G'\left(\underbrace{\frac{w + \lambda z}{\sqrt{1 + \lambda^2}} \cdot x}_{w_1}\right) G'\left(\underbrace{\frac{w + \lambda z'}{\sqrt{1 + \lambda^2}} \cdot x}_{w_2}\right) \ell(v \cdot x)\right]\right).$$

By expanding $G'$ and $\ell$, we get a Gaussian expectation of a triple product of Hermite polynomials, whose scaling can be computed thanks to Lemma 12, recalling that $G$ is even and then $c_0^{G'} = 0$. In particular, thanks to Corollary 13, we have

$$\mathbb{E}_{\mathbb{P}_0}[G'(w_1 \cdot x)G'(w_2 \cdot x)\ell(v \cdot x)] = \Theta\left((w_1 \cdot w_2)^{k_2^* - 1}\right)$$

Since $z \cdot z' = \Theta(1/\sqrt{d})$, we can avoid the dependence on the replicas by simply choosing $\lambda \ll d^{1/4}$. This choice corresponds to Equation (15) in Appendix E. Hence, $w_1 \cdot w_2 = \Theta(\lambda^{-2})$. We can conclude that the noise scales as

$$\mathbb{E}[\|g\|^2] = \Theta(d\lambda^{-2k_2^*}).$$

By combining signal and noise, we have that

$$\text{SNR} = \frac{\alpha^2}{\lambda^4} d^{-(k_1^* - 1)} \frac{1}{d\lambda^{-2k_2^*}} = \alpha^2 \frac{\lambda^{2(k_2^* - 2)}}{d^{k_1^* - 1}}.$$

This implies that, in the case of the optimal $\lambda = d^{1/4}$, the ODE describing the behavior of the overlap $\alpha$ for small $t \geq 0$ is nothing but Equation (9).

Therefore, the number of data points required to reach an order-one overlap is $d^{k^*/2}$, i.e. the optimal one according to the fundamental limits of recovery, if and only if $k_1^* = k_2^*$ and than we have heuristically derived the results of Theorem 4. The following theorem, which we prove in Appendix E, makes rigorous the heuristics. It holds for any input model satisfying Assumption 1, Assumption 2, and the following Assumption 3:

**Assumption 3.** *We assume $\ell \in \text{L}^\infty(\mathbb{R}^d, \mathbb{P}_0)$ and that the loss (2) is strictly monotonically increasing in $\alpha = w \cdot v$.*

**Theorem 4** (Escaping mediocrity). *Define $k_1^*$ and $k_2^*$ as the information exponents of the population loss (2) and the contrast function, respectively. Assume that $w_0 \sim Unif(\mathbb{S}^{d-1})$ and that the initial overlap $\alpha_0 := w_0 \cdot v$ is such that $\alpha_0 \gtrsim 1/\sqrt{d}$. Consider $\lambda \in [1, d^{1/4}]$. Then, there exist $\eta \in \mathbb{R}$ and $n \in \mathbb{N}$, satisfying*

$$\begin{cases} n = O\left(d^{k_1^* - 1} \lambda^{-2(k_2^* - 2)} \text{polylog}(d)\right), \\ \eta = O\left(d^{-k_1^*/2} \lambda^{2(k_2^* - 1)} \text{polylog}(d)\right) \end{cases}$$

*such that $\alpha_n \geq 1 - d^{-1/4}$ with high probability. Here, $\alpha_n := w_n \cdot v$ and $w_n$ is the estimator given by smoothed online SGD with $\eta_n = \eta$ and $\lambda_n = \lambda$.*

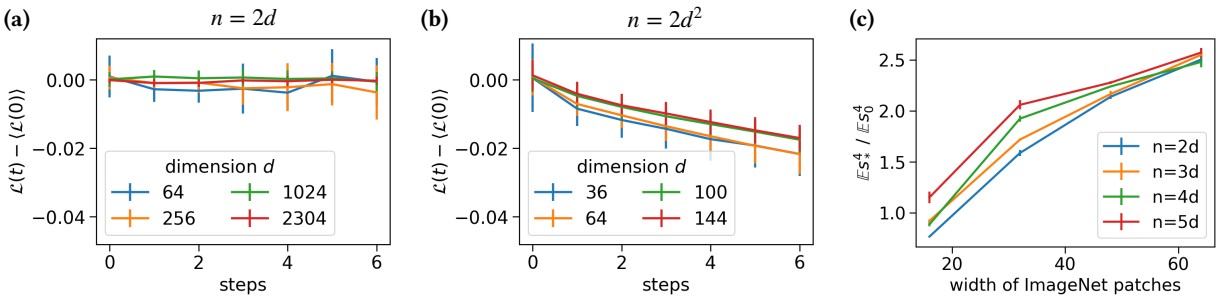

*Figure 4.* **Performance of FastICA on ImageNet.** We plot the difference between the test loss $\mathcal{L}(t)$ at step $t$, and the test loss $\langle\mathcal{L}(0)\rangle$ at initialisation averaged over twenty initialisations. At linear sample complexity **(a)**, the test loss remains close to its initial values, but it clearly decreases at quadratic sample complexity **(b)**. **(c)** Ratio of the fourth moment of projections $s_*$ of ImageNet patches of the given width along a Gabor filter obtained via FastICA, and of the fourth moment of projections $s_0$ along a random direction. Different colours refer to the size of the training set from which the Gabor filters were obtained. Full experimental details in Appendix A.4.

## 4. The search phase of FastICA on images

We finally investigate whether FastICA also exhibits an extended search phase on real data. To that end, we ran FastICA on patches sampled at random from the ImageNet data set (Deng et al., 2009). In Figure 4, we show the difference between the test loss $\mathcal{L}(t)$ at step $t$ of FastICA and the loss $\langle\mathcal{L}(0)\rangle$ averaged over twenty randomly chosen initial conditions. We find that at linear sample complexity when $n = 2d$, Figure 4(a), FastICA is clearly stuck in a search phase: the loss does not deviate significantly from its value at initialisation. Meanwhile in the quadratic regime at $n = 2d^2$, Figure 4(b), FastICA does recover a non-trivial Gaussian direction from the start, faster than our theory would suggest.

Where does this non-Gaussian signal come from that FastICA picks up in the images? A natural idea would be to suggest that it comes from the third-order cumulant, which is non-zero for real images in contrast to the spiked cumulant with Rademacher latent $\nu$. The corresponding information exponent $k^* = 3$ would then suggest that $n \asymp d^{k^*-1}$ is the transitionary regime, where recovery is possible in finite dimensions. However, since the contrast functions used in FastICA are symmetric, FastICA is blind to information carried by odd cumulants. We demonstrate this explicitly by constructing a latent distribution which has mean zero, unit variance, a non-trivial third-order cumulant, but zero fourth-order cumulant, and find that FastICA does not pick up any signal, see Figure 8. Another potential explanation is that the latent variables of images are super-exponential, and follow an approximate Laplace distribution (Wainwright & Simoncelli, 1999). However, we verified experimentally that when the latent variable $\nu^\mu$ in the noisy ICA model has a Laplace distribution, Theorem 2 still accurately describes the sample complexity required for recovery, see Figure 9. However, the longer tail induces stronger finite-size fluctuations, which might contribute to the success of FastICA at quadratic sample complexities.

Indeed, we found strong finite-size effects when attempting to measure the effective signal-to-noise ratio of the non-Gaussian directions in the images. In the spiked cumulant model, an empirical measure of signal-to-noise ratio can be obtained by dividing the fourth moment of the projection of inputs along the spike, $s_* = v \cdot x$, with the fourth moment of inputs projected along a random direction $s_0 = w_0 \cdot x$. As a surrogate for the "spike" in real images, we ran FastICA to obtain clean Gabor filters from $n = 2, 3, 4, 5d$ ImageNet patches (as detailed in Appendix A.2) and computed the ratio of the fourth moment of projections $s_*$ along these filters with projections $s_0$ along random directions. For the spiked cumulant with standard Laplace prior, this ratio is equal to 2. In Figure 4(c), we see that this ratio for images tends towards an larger value, but we note that the empirical estimate only converges slowly with width, hinting at important finite-size fluctuations.

## 5. Concluding perspectives

Despite their simplicity, Independent Component Analysis yields similar filters as the early layers of deep convolutional neural networks, and therefore offers an interesting model to study feature learning from non-Gaussian inputs. Here, we considered the problem of recovering a single feature encoded in the non-Gaussian input fluctuations and established almost sharp sample complexity thresholds for FastICA and SGD. Our analysis revealed the poor sample complexity of FastICA, which might explain some of its problems in high-dimensional settings, while our experiments suggest that images have strong enough non-Gaussian features to compensate for this. Going forward, it will be intriguing to extend our analysis to models with several non-Gaussian directions, along the lines of Dandi et al. (2024) and Bardone & Goldt (2024). Finally, developing a model for synthetic data from which localised, oriented filters can be learnt looms as an intriguing challenge (Ingrosso & Goldt, 2022; Lufkin et al., 2024).

## Impact statement

This paper presents work whose goal is to advance the field of Machine Learning. There are many potential societal consequences of our work, none which we feel must be specifically highlighted here.

## Acknowledgements

We thank Mitya Chklovskii for valuable discussions. SG acknowledges funding from the European Research Council (ERC) under the European Union's Horizon 2020 research and innovation programme, Grant agreement ID 101166056, and funding from the European Union–NextGenerationEU, in the framework of the PRIN Project SELF-MADE (code 2022E3WYTY – CUP G53D23000780001). SG and FR acknowledge funding from Next Generation EU, in the context of the National Recovery and Resilience Plan, Investment PE1 – Project FAIR "Future Artificial Intelligence Research" (CUP G53C22000440006).

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

# A. Experimental details

In this appendix, we collect detailed information on how we ran the various experiments of this paper.

## A.1. Figure 1

**(a)** We plot the filters of the first convolutional layer of an AlexNet trained on ImageNet using mini-batch size of 128, momentum of 0.9, weight decay of $10^{-4}$, cosine learning rate schedule with initial learning rate of $10^{-2}$. We also trained a DenseNet121 (Huang et al., 2017) and a ResNet18 (He et al., 2016) using the same recipe, but with larger initial learning rate $10^{-1}$.

**(b)** The Gabor filters were obtained with online SGD performed on 100 000 patches of size $64 \times 64$ extracted from ImageNet. We have decided to perform ICA with SGD to use the typical algorithm employed to train deep neural networks. We perform PCA before running ICA by keeping 600 principal components. The contrast function was $G(s) = -\exp(-s^2/2)$.

**(c)** We applied principal component analysis to the same ImageNet patches to obtain the leading principal components, and plot the eight leading ones from left to right, and top to bottom.

**(d)** For all three networks trained on ImageNet, we took 50 logarithmically spaced snapshots of the weights during training. For each snapshot, we extracted the weights for each first-layer convolution and computed the dot-products $s_\mu^k = w^k \cdot x^\mu$ between the weight of the $k$th convolution for randomly sampled patches from ImageNet $x^\mu$. We normalised the $s_\mu^k$ for each neuron (fixed $k$) to have variance 1, and computed the excess kurtosis of the resulting $s_\mu^k$. We then averaged over all the neurons to obtain the curves shown in Figure 1(b).

**(e)** The snapshots were taken from one of the AlexNet models.

## A.2. Figure 2

Here we show the performance of FastICA on the spike cumulant model (5) in three different regimes, in the case of $d = 50$ and $\beta = 15$. The spike $v$ has been sampled uniformly in the sphere $\mathbb{S}^{d-1}$. The regimes correspond to the batch-sizes of $n = d^2, n = d^{3+\delta}$ and $n = d^4$, respectively, for $\delta = 0.2$. Each batch contains $n$ fresh data points sampled from the distribution of the inputs $\mathbb{P}$. The contrast function is $G(s) = -\exp(-s^2/2)$. We display on the $y$-axis the overlap $w_t \cdot v$ for $t = 0, \ldots, 4$ (on the $x$-axis), where $w_t$ is the updated weight vector provided by FastICA in the first four step, plus initialisation $w_0 \sim \mathbb{S}^{d-1}$ under the constraint that $w_0 \cdot v = 1/\sqrt{d}$. An average over 15 runs has been taken.

## A.3. Figure 3

Here we show the velocity for recovering the spike of smoothed online SGD in three different regimes: linear, quadratic and cubic. The noisy ICA model is considered. We use $d = 40$, $\beta = 15$ and a total number of data points of $n = 5d^3$. The parameter $\lambda$ is the optimal one, i.e. $\lambda = d^{1/4}$. The learning rate is $\eta = 2.9d^{(-k_1^*/2)}\lambda^{(2k_2^*-2)}$, where $k_1^* = 4$ and $k_2^* = 2$, since $G(s) = -\exp(-s^2/2)$. The average is made over 30 runs.

## A.4. Figure 4

**(a) and (b)** We perform ICA on patches of ImageNet images with increasing dimensions, and randomly drawn. The chosen contrast function is $G(s) = -\exp(-s^2/2)$. The standard centering and whitening pre-processing procedure is performed before running the algorithm.

**(c)** We first ran FastICA on $n = 2, 3, 4, 5d$ ImageNet image patches to obtain clean Gabor filters, which we used as surrogates of the spike $v$. To that end, we used the "canonical preprocessing" of Hyvärinen et al. (2009), meaning we whitened inputs and ran the FastICA algorithm only in the subspace spanned by the top $k$ principal components (here, $k = 50$ for all widths). We then computed the fourth moment of projections $s_*$ of an independent test set of ImageNet images along these Gabor filters, and divided this by the fourth moment of projections along random directions $s_0$.

## A.5. The "3not4" prior on the latent variable, Figure 8

For this experiment, we run online FastICA as in Figure 2 on a noisy spiked cumulant model, but with a prior over the latent variable that has mean zero, unit variance, a non-trivial third-order cumulant, but zero fourth-order cumulant. More

specifically, we consider a latent $\nu$ that can take the values $-1, 0$ and $2$ with probabilities $1/3, 1/2$ and $1/6$.

## B. Properties and definitions

### B.1. Notation

Throughout we use the standard notations $o, O, \Theta, \Omega, \omega$ for asymptotics. We recall them here for sequences, but they generalize in the straight forward way in the case of functions.

Let $\{a_k\}_{k\in\mathbb{N}}$, $\{b_k\}_{k\in\mathbb{N}}$ be two real valued sequences. Then:

$$a_k \in o(b_k) \iff \lim_{k\to\infty} \frac{a_k}{b_k} = 0$$

$$a_k \in O(b_k) \iff \exists\, C > 0 \in \mathbb{R} \quad \forall k > k_0 \quad |a_k| \le C|b_k|$$

$$a_k \in \Theta(b_k) \iff \exists\, C_1, C_2 > 0 \quad \forall k > k_0 \quad C_1 b_k \le a_k \le C_2 b_k$$

$$a_k \in \Omega(b_k) \iff \exists\, C > 0 \in \mathbb{R} \quad \forall k > k_0 \quad |a_k| \ge C|b_k|$$

$$a_k \in \omega(b_k) \iff \lim_{k\to\infty} \frac{|a_k|}{|b_k|} = \infty$$

Occasionally we use the shorthands $a_n \ll b_n$ for $a_n = o(b_n)$, $a_n \lesssim b_n$ for $a_n = O(b_n)$ and $a_n \asymp b_n$ for $a_n = \Theta(b_n)$.

### B.2. Probability notions

We recall the properties of the space of squared integrable functions with respect to the standard normal distribution $\mathbb{P}_0$:

**Definition 5** (Gaussian scalar product). *Given $f, g : \mathbb{R}^d \to \mathbb{R}$, we define the $\mathrm{L}^2$-scalar product as*

$$\langle f, g \rangle_{\mathbb{P}_0} := \mathbb{E}_{x\sim\mathbb{P}_0}\left[f(x)g(x)\right].$$

*The space $\mathrm{L}^2(\mathbb{R}^d, \mathbb{P}_0)$ contains all the measurable functions $f : \mathbb{R}^d \to \mathbb{R}$ such that $||f||^2 := \langle f, f \rangle_{\mathbb{P}_0} < \infty$.*

Note that this space with the product defined above is an Hilbert space, and the set of the Hermite polynomials (see next section) is an orthogonal basis.

**Definition 6** (Sub-Gaussianity). *Let $Z$ be a real valued random variable, we say that $Z$ is sub-Gaussian if it has finite sub-Gaussian norm, defined as:*

$$||Z||_{\psi/2} := \inf\left\{t > 0 \mid \mathbb{E}[\exp\left(Z^2/t^2\right)] \le 2\right\}$$

For more details on sub-Gaussian distributions we refer to (Vershynin, 2018), Section 2.5.

We also need to introduce the space $\mathrm{L}^\infty(\mathbb{R}^d, \mathbb{P}_0)$ as the space of essentially bounded functions, with finite $|| \cdot ||_\infty$ norm, defined as:

$$||f||_\infty := \inf\left\{C > 0 \mid \mathbb{P}_0(|f(x)| \le C) = 1\right\}$$

Note that the likelihood ratio $\ell = \frac{d\mathbb{P}}{d\mathbb{P}_0}$ for model (5) belongs to this space as soon as the tails of $\nu$ decrease faster than a standard Gaussian. For instance all the distributions with bounded support are included.

We often refer to the notion of events that happen *with high probability*, defined as follows.

**Definition 7** (Events happening with high probability events). *The sequence of events $\{E_d\}_{d\in\mathbb{N}}$ happens with* high probability *if for every $k \ge 0$, there exists $d_k$ such that for $d \ge d_k$*

$$1 - P(E_d) \le \frac{1}{d^k}$$

### B.3. Hermite polynomials

In this section we group some facts about Hermite polynomials that will be needed for the subsequent proofs. For a comprehensive overview on the topic we refer to (McCullagh, 2018).

**Definition 8** (Hermite expansion). *Consider a function $f : \mathbb{R} \to \mathbb{R}$ that is square integrable with respect to the standard normal distribution $p(x) = (1/\sqrt{2\pi})\, e^{-x^2/2}$. Then there exists a unique sequence of real numbers $\{c_k\}_{k \in \mathbb{N}}$ called Hermite coefficients such that*

$$f(x) = \sum_{k=0}^{\infty} \frac{c_k}{k!} h_k(x) \quad and \quad c_k(x) := \mathbb{E}_{x \sim \mathcal{N}(0,1)}[f(x) h_k(x)],$$

*where $h_i$ is the $i$-th probabilist's Hermite polynomial.*

**Definition 9** (Information exponent). *Given any function $f$ for which the Hermite expansion exists, its information exponent $k^* = k^*(f)$ is the smallest index $k \geq 1$ such that $c_k \neq 0$.*

**Lemma 10** (Expectations of Triple and Quadruple products). *For any $i, j, k \in \mathbb{N}$, we have*

$$I_3^*(i, j, k) := \mathbb{E}_{x \sim \mathcal{N}(0,1)}[h_i(x) h_j(x) h_k(x)] = \frac{i!\, j!\, k!}{\left(\frac{i+j-k}{2}\right)! \left(\frac{i+k-j}{2}\right)! \left(\frac{j+k-i}{2}\right)!}$$

*if the denominator exists, that is if $i + j + k$ is even and the sum of any two of $i, j$ and $k$ is strictly less than the third, or zero otherwise. What's more, fix the indices $m_1, m_2, m_3, m_4 \in \mathbb{N}$ and order them such that $m_1 \geq m_2 \geq m_3 \geq m_4$. Define $M := (m_1 + m_2 + m_3 + m_4)/2$.*
*Then, given $m := \min(m_3, m_4)$ and $I_4^*(m_1, m_2, m_3, m_4) := \mathbb{E}_{x \sim \mathcal{N}(0, \mathbb{1}_d)}[h_{m_1}(x) h_{m_2}(x) h_{m_3}(x) h_{m_4}(x)]$, we have*

$$I_4^*(m_1, m_2, m_3, m_4) = \sum_{\nu=0}^{m} \frac{(m_3 + m_4 - 2\nu)!\, m_1!\, m_2!\, m_3!\, m_4}{(M - m_3 - m_4 - \nu)!(M - m_1 - \nu)!(M - m_2 - \nu)!(m_3 - \nu)!(m_4 - \nu)!\nu!}$$

*if $M \in 2\mathbb{N}$ and the denominator exists. Otherwise, $I_4^*(m_1, m_2, m_3, m_4) = 0$.*

**Lemma 11.** *Consider $w_1, w_2 \in \mathbb{S}^{d-1}$ and set $\alpha = w_1 \cdot w_2$. Then, for any $i, j \in \mathbb{N}$*

$$\mathbb{E}_{x \sim \mathcal{N}(0, \mathbb{1}_d)}[h_i(w_1 \cdot x) h_j(w_2 \cdot x)] = i!\, \alpha^i\, \delta_{ij}.$$

To extend this orthogonality property to triple an quadruple products of Hermite polynomials, we use the same technique as in (Buet-Golfouse, 2015)

**Lemma 12** (Orthogonality property). *Fix $i, j, k \in \mathbb{N}$. Set $I_3 := \mathbb{E}_{x \sim \mathcal{N}(0, \mathbb{1}_d)}[h_i(w_1 \cdot x) h_j(w_2 \cdot x) h_k(w_3 \cdot x)]$. Consider three different spikes $w_1, w_2, w_3 \in \mathbb{S}^{d-1}$ and set $\rho := w_1 \cdot w_2, \tau := w_1 \cdot w_3$ and $\eta := w_2 \cdot w_3$. Then, we have*

$$I_3 = \sum_{\ell=0}^{j} \frac{i!j!k!}{(j-\ell)!^2 \left(\frac{i+j-k}{2}\right)! \left(\frac{i+k-j}{2}\right)! \left(\frac{k-j+2\ell-i}{2}\right)!} \rho^\ell \tau^{k-j+\ell} (\eta - \tau\rho)^{j-\ell}$$

*if the denominator exists, and $I_3 = 0$ otherwise.*

*Proof.* We start by writing $w_2$ as sum of its projection along the first spike $w_1$ and its part in the orthogonal space, that is $w_2 = \rho w_1 + \sqrt{1 - \rho^2} u$, where $u \in \mathbb{S}^{d-1}$ and $u \cdot w_1 = 0$. Then, clearly, $w_2 \cdot x = \rho y_1 + \sqrt{1 - \rho^2} y_u$, where $y_1 := w_1 \cdot x$ and $y_u := u \cdot x$.
We can do the same for the third spike, that is $w_3 = \tau w_1 + \sqrt{1 - \tau^2} u'$, where $u' \in \mathbb{S}^{d-1}$ and $u' \cdot w_1 = 0$. Define $y_{u'} = u' \cdot x$.
We can now exploit the fact that any Hermite polynomial of a sum can be written as the sum of Hermite polynomials in the following way, that is for any $m \in \mathbb{N}$ we have that $h_m(\alpha z_1 + \sqrt{1 - \alpha^2} z_2) = \sum_{n=0}^{m} \binom{m}{n} \alpha^n (\sqrt{1 - \alpha^2})^{m-n} h_\ell(z_1) h_{m-n}(z_2)$, for any $\alpha \in [0, 1]$. By applying this formula to $w_2$ and $w_3$, we get

$$I_3 = \sum_{\ell, \ell'=0}^{j,k} \binom{j}{\ell} \binom{k}{\ell'} \rho^\ell \tau^{\ell'} (\sqrt{1 - \rho^2})^{j-\ell} (\sqrt{1 - \tau^2})^{k-\ell'} \underbrace{\mathbb{E}[h_i(y_1) h_\ell(y_1) h_{\ell'}(y_1)]}_{(1)} \underbrace{\mathbb{E}[h_{j-\ell}(y_u) h_{k-\ell'}(y_{u'})]}_{(2)}$$

where we have used the fact that $y_1$ is independent from $y_u$ and $y_{u'}$ since $w_1 \cdot u = w_2 \cdot u' = 0$. Then, in view of Lemma 10 and Lemma 11, we have

$$(1) = \frac{i!\ell!\ell'!}{\left(\frac{i+\ell-\ell'}{2}\right)! \left(\frac{i+\ell'-\ell}{2}\right)! \left(\frac{\ell+\ell'-i}{2}\right)!} \quad and \quad (2) = (j-\ell)!(u \cdot u')^{j-\ell} \delta_{j-\ell=k-\ell}.$$

Hence, imposing that $\ell' = k - j - \ell$, we obtain

$$I_3 = \sum_{\ell=0}^{j} \frac{i!j!k!}{(j-\ell)!\left(\frac{i+j-k}{2}\right)!\left(\frac{i+k-j}{2}\right)!\left(\frac{k-j+2\ell-i}{2}\right)!} \rho^\ell \tau^{k-j+\ell}[\sqrt{1-\rho^2}\,\sqrt{1-\tau^2}\,(u\cdot u')]^{j-\ell}$$

when the denominator exists and it vanishes otherwise. Therefore, the thesis follows from the fact that we have $u \cdot u' = \dfrac{\eta - \tau\rho}{\sqrt{1-\tau^2}\sqrt{1-\rho^2}}.$

$\square$

**Corollary 13.** *Assume that $\ell$ is a probability distribution and that $G \in C^1(\mathbb{R})$ is even. Then, if $w_1, w_2, w_3 \in \mathbb{S}^{d-1}$ and $w_1 \neq w_2 \neq w_3$, we obtain that*

$$E_{x\sim\mathcal{N}(0,\mathbb{1}_d)}[G'(w_1\cdot x)G'(w_2\cdot x)\ell(w_3\cdot x)] = \Theta(w_1\cdot w_2^{k_2^*-1}).$$

*Proof.* By considering the Hermite expansion of $G'$ and $\ell$, it clearly follows that

$$E_{x\sim\mathcal{N}(0,\mathbb{1}_d)}[G'(w_1\cdot x)G'(w_2\cdot x)\ell(w_3\cdot x)] = \sum_{i,j,k=0}^{\infty} \frac{c_i^{G'} c_j^{G'} c_k^{\ell}}{i!j!k!} E_{x\sim\mathcal{N}(0,\mathbb{1}_d)} \underbrace{[h_i(w_1\cdot x)h_j(w_2\cdot x)h_k(w_3\cdot)]}_{(\star)}, \qquad (10)$$

where, in view of Lemma 12, when $\rho := w_1 \cdot w_2$, $\tau := w_1 \cdot w_3$ and $\eta := w_2 \cdot w_3$, it holds that

$$\mathbb{E}[(\star)] = \sum_{t=0}^{j} \frac{i!j!k!}{(j-t)!^2\left(\frac{i+j-k}{2}\right)!\left(\frac{i+k-j}{2}\right)!\left(\frac{k-j+2t-i}{2}\right)!} \rho^t \tau^{k-j+t}(\eta-\tau\rho)^{j-t} \qquad (11)$$

and then

$$\underset{x\sim\mathcal{N}(0,\mathbb{1}_d)}{\mathbb{E}}[G'(w_1\cdot x)G'(w_2\cdot x)\ell(w_3\cdot x)] = \sum_{i,j,k=0}^{\infty}\sum_{t=0}^{j} \text{coeff } \rho^t \tau^{k-j+t}(\eta-\tau\rho)^{j-t},$$

where

$$\text{coeff} = \frac{c_i^{G'} c_j^{G'} c_k^{\ell}}{(j-t)!^2\left(\frac{i+j-k}{2}\right)!\left(\frac{i+k-j}{2}\right)!\left(\frac{k-j+2t-i}{2}\right)!}.$$

We can compute the way this sum scales with the dimension by recalling that since $\ell$ is a probability distribution, we have $c_0^\ell = 1$. However, since $G$ is even, $c_0^{G'} = c_1^G = 0$. The smallest indices such that the sum does not vanish are then $k = 0$ and $i = j = k_2^* - 1$. Then, it turns out that the only term which survives in the sum when $k = 0$ is the one with $t = k_2^* - 1$. Therefore, up to checking that the other terms in the sum are dominated by the addendum corresponding to $k = 0$, $i, j = k_2^* - 1$ and $t = k_2^* - 1$, we have

$$\underset{x\sim\mathcal{N}(0,\mathbb{1}_d)}{\mathbb{E}}[G'(w_1\cdot x)G'(w_2\cdot x)\ell(w_3\cdot x)] = \Theta(\rho^{k_2^*-1}).$$

$\square$

**Corollary 14** (used formulas with same argument)**.** *Fix $k = 1$ and $i, j \in \mathbb{N}$. Consider $I_3$ given by 12 in the case of $w_3 = w_2$. Then,*

$$I_3 = \mathbb{E}_{x\sim\mathcal{N}(0,1)}[h_i(w_1)h_j(w_2)h_1(w_1)] = \frac{i!j!}{\left(\frac{i+j-1}{2}\right)!\left(\frac{i+1-j}{2}\right)!\left(\frac{j+1-i}{2}\right)!} \rho^j,$$

*when the denominator exists and $I_3 = 0$ otherwise.*
*Moreover, for any $i, j, k, t \in \mathbb{N}$, consider $I_4 := \mathbb{E}_{x\sim\mathcal{N}(0,1)}[h_i(w_1\cdot x)h_j(w_1\cdot x)h_k(w_2\cdot x)h_t(w_3\cdot x)]$, with $w_1, w_2, w_3 \in \mathbb{S}^{d-1}$. Then,*

$$I_4 = \sum_{\ell=0}^{k}(-1)^{k-\ell}\binom{k}{\ell}\binom{t}{t-k+\ell}\rho^k\tau^t I_4^*(i,j,\ell,t-k+\ell),$$

*where $I_4^*$ is defined in Lemma 10 and the indeces $i, j, \ell$ and $t - k + \ell$ are ordered.*

*Proof.* The first part is obtained by repeating the proof of Lemma 12 with $w_2 = \rho w_1 + \sqrt{1 - \rho^2} u$, for $u \in \mathbb{S}^{d-1}$ and $u \cdot w_1 = 0$, and using the fact that $\mathbb{E}_{x \sim \mathcal{N}(0, \mathbb{1}_d)}[h_{j-\ell}(u \cdot x)] = \delta_{j-\ell}$.

For the second part, it is sufficient to repeat the usual procedure by decomposing both $w_2$ and $w_3$ such that $w_2 = \rho w_1 + \sqrt{1 - \rho^2} u$ and $w_3 = \tau w_1 + \sqrt{1 - \tau^2} u'$ for $u, u' \in \mathbb{S}^{d-1}$ and $w_1 \cdot w_2 = w_1 \cdot w_3 = 0$. Then, the thesis follows from the second part of Lemma 12. $\qquad \square$

## C. Measures of non-Gaussianity for ICA

Whether the population loss has to be (globally) maximized or minimized with respect to the weight vector $w \in \mathbb{S}^{d-1}$ depends on the data distribution and the choice of the contrast function $G$. Therefore, we briefly comment on this, by referring to (Hyvärinen & Oja, 2000) and (Hyvärinen et al., 2001) for more details. What does maximising *non-Gaussianity* mean? Equivalently: what is a suitable measure for non-Gaussianity? A standard approach is to choose the *negentropy* as a measure of the non-Gaussianity of the random vector $x \sim \mathbb{P}$, i.e.

$$J(x) := H(x_{Gauss}) - H(x),$$

where $H(x) := -\mathbb{E}_{\mathbb{P}}[\log(x)]$ and $x_{Gauss} \sim \mathbb{P}_0$. We assume that $x$ has zero mean and that its covariance matrix is the identity. The negentropy, being a Kullback-Leibler divergence, is always non-negative. However, to compute $J$ one needs to access the density of the random vector $x$, which is usually unknown and not easy to approximate. Hence, the classical approach is to approximate negentropy indirectly. In Section 5 of (Hyvärinen et al., 2001), the authors show that the negentropy can be approximated by

$$\tilde{\mathcal{L}}(w) = \left( \mathbb{E}\left[ G(w \cdot x) \right] - \mathbb{E}\left[ G(x_{Gauss}) \right] \right)^2$$

where $G$ is even and does not grow too fast because of reasons of numerical stability. Typically, $G$ is one of the contrast functions from Equation (3). Note that this optimization problem is slightly different from Equation (1) (which is the setting considered in practice, as mentioned in (Hyvärinen & Oja, 2000)). The reason why it is convenient to remove the square is clear as soon as the Hermite expansion Equation (14) is performed, together with the fact $\mathbb{E}\left[ G(x_{Gauss}) \right] = c_0^G c_0^\ell$:

$$\begin{aligned}
\tilde{\mathcal{L}}(w) &= \left( \mathcal{L}(w) - \mathbb{E}\left[ G(x_{Gauss}) \right] \right)^2 \\
&= \left( c_0^G c_0^\ell + \frac{c_{k^*}^G c_{k^*}^\ell}{k!} \alpha^{k^*} + o(\alpha^{k^*}) - c_0^G c_0^\ell \right)^2 \\
&= \left( \frac{c_{k^*}^G c_{k^*}^\ell}{k!} \right)^2 \alpha^{2k^*} + o(\alpha^{2k^*})
\end{aligned}$$

hence the information exponent of this loss function is doubled with respect to $\mathcal{L}$ worsening the sample complexity for the search phase (geometrically, this can be seen as the square "flattening" the loss around the origin).

The removal of the square comes with the side effect that it is not sufficient to just minimize the loss, since the sign becomes relevant: $G$ needs to me maximised in case of platykurtic data since the cumulants of the data are smaller than the gaussian ones, so $\mathbb{E}[G(w \cdot x)] - \mathbb{E}[G(x_{gauss})] < 0$, and conversely needs to be minimized in case of leptokurtic data distributions.

### C.1. Details on the noisy ICA model model

Here we expand the population loss (2) in the specific case where the inputs are distributed according to the spike cumulant model (5). We can expand the likelihood ratio $\ell$ in series of Hermite polynomials in the sense of Definition 8, and write

$$\ell(v \cdot x) = \sum_{j=0}^{\infty} \frac{c_j^\ell}{j!} h_j(v \cdot x).$$

Note that due to the symmetry of $\nu$, $c_{2j+1}^\ell = 0$ for any $j \in \mathbb{N}$. Moreover, due to the whitening matrix, also $c_2^\ell = 0$; so, apart from $c_0^\ell = 1$, the smallest non-zero coefficient is $c_4^\ell$, the one corresponding to first non-trivial cumulant, i.e. the kurtosis.

Using the orthogonality identity for Hermite polynomials (Lemma 11 in Appendix B.3), we get

$$
\mathcal{L}(w) = \mathbb{E}_{\mathbb{P}_0}[G(w \cdot x)\,\ell(v \cdot x)]
$$

$$
= \sum_{i,j \geq 0}^{\infty} \frac{c_i^G c_j^\ell}{i!j!}\, \mathbb{E}_{\mathbb{P}_0}[h_i(w \cdot x)h_j(v \cdot x)]
$$

$$
= c_0^G c_0^\ell + \frac{c_4^G c_4^\ell}{4!}\alpha^4 + o(\alpha^4)
$$

where $\alpha = w \cdot v$ is the only order parameter of the system. For this calculation, we have assumed that $c_4^G \neq 0$, which holds for the contrast functions in Equation (3). This implies that the inference problem of detecting the spike $v$ has information exponent $k^* = 4$, in the sense of Definition 8 in Appendix B.3.

Note also that, for this specific data distribution, $\ell$ can be computed explicitly, that is

$$
\ell(y) = \mathbb{E}_\nu \left[ \sqrt{1+\beta} \exp\left( -\frac{1+\beta}{2}\left(\nu - \frac{\beta}{1+\beta}v\right)^2 + \frac{1}{2} \right) \right]. \tag{12}
$$

For more details regarding this computation, we refer to Szekely et al. (2024).

# D. FastICA

In Figure 5 we show numerically that the large fluctuations seen in Figure 2 (middle) hold for various values of $\epsilon$. We run FastICA on the noisy ICA model with size batch $d^{3+\epsilon}$, for $\epsilon = 0.2, 0.5, 0.8$. We plot $\alpha_v = v \cdot w$ in the first four iterations, plus the overlap at initialisation.

We prove now Proposition 1, which guarantees that the second order term in the iteration of FastICA does not only speed up the convergence of the algorithm, but helps in escaping the search phase too: indeed, when it is not present, the recovered signal after the first giant step scales as the random weight initialisation, i.e. $\alpha_1^2 = (v \cdot w_1)^2 = O(1/d)$. However, when the iteration of FastICA is complete, the relevant direction is learned in one single step, if the size-batch is infinite and the dimension of the inputs diverge.

*Proof of Proposition 1 (Infinite batch size).* We start by recalling that, up to normalisation, the first iteration of FastICA in the population limit, i.e. for a batch of infinite size, reads

$$
\widetilde{w}_1 = \mathbb{E}[x\, G'(w_0 \cdot x)] - \mathbb{E}[G''(w_0 \cdot x)]\, w_0,
$$

where the weight vector at initialisation $w_0$ is uniformly drawn in the unit sphere in $\mathbb{R}^d$. Here, we have considered the complete iteration, with both the first order and the second order terms. Since we assume we are in a high dimensional regime, we have that the overlap between the initialisation and the spike that we want to recover scales as $\alpha_0 := w_0 \cdot v = \Theta(1/\sqrt{d})$. We want to prove that, after the first step, the spike has been totally learned when $d \to +\infty$, namely

$$
\alpha_1^2 := (w_1 \cdot v)^2 = 1 - o(1).
$$

To do so, we can expand in the series of Hermite polynomials the squared overlap

$$
\alpha_1^2 = \frac{\pi_1^2}{\|\widetilde{w}_1\|}, \text{ with } \pi_1 := \widetilde{w}_1 \cdot v \tag{13}
$$

at a sufficiently high order. First of all, we take into account the first and the second order terms $f := x\, G'(w_0 \cdot x)$ and $g := G''(w_0 \cdot x)w_0$ and compute their expectations with respect to the data distribution.

We compute $\mathbb{E}[f]$ by noticing that $f \in \mathbb{R}^d$ can be written as the sum of a spherical gradient and a derivative along the direction of the initialisation, in the following sense: if $L(w, x) := G(w \cdot x)$, we have that

$$
f = (\mathbb{1}_d - w_0 w_0^\top)\nabla_w L(w,x)\Big|_{w=w_0} + w_0 w_0^\top \nabla_w L(w,x)\Big|_{w=w_0} = \underbrace{\nabla_{\text{sph}} L(w,x)\Big|_{w=w_0}}_{(\star)_1} + \underbrace{w_0 w_0^\top \nabla_w L(w,x)\Big|_{w=w_0}}_{(\star)_2}.
$$

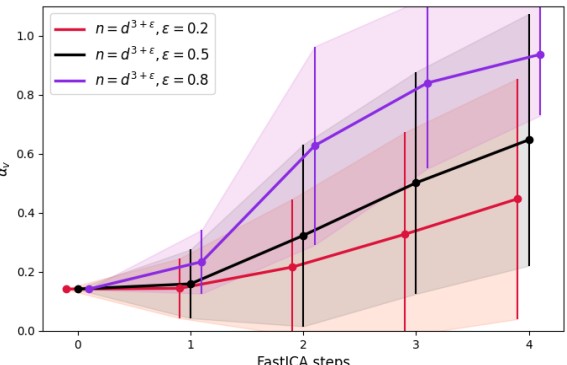

*Figure 5.* **Performance of FastICA on the spiked cumulant model for various sample complexity regimes.** We run the standard FastICA updates (4) on data drawn from a noisy ICA model (5). At each step of the algorithm, we draw a new batch of size $n = d^{3+\varepsilon}$ with $\varepsilon = 0.2, 0.5, 0.8$. We plot the overlap between the planted non-Gaussian direction $v$ and the estimate $w$ produced by FastICA. Theorem 2 confirms this picture for a general data model.

This splitting has to be done since we need unit vectors to be allowed to apply the orthogonality formulas in Section B. By using the likelihood ratio trick, we get

$$
\begin{aligned}
\mathbb{E}[(\star)_1] =& \mathbb{E}\left[\left.\frac{\partial}{\partial w}G(w \cdot x)\right|_{w=w_0}\right] = \mathbb{E}_{x\sim\mathcal{N}(0,\mathbb{1}_d)}\left[\left.\frac{\partial}{\partial w}G(w \cdot x)\right|_{w=w_0}\ell(v \cdot x)\right] \\
=& \frac{\partial}{\partial w}\mathbb{E}_{x\sim\mathcal{N}(0,\mathbb{1}_d)}\left[\left.\left(\sum_{i=0}^{\infty}\frac{c_i^G}{i!}h_i(w \cdot x)\right)\left(\sum_{j=0}^{\infty}\frac{c_j^\ell}{j!}h_j(w \cdot x)\right)\right|_{w=w_0}\right] \\
=& \frac{\partial}{\partial w}\sum_{i,j=0}^{\infty}\frac{c_i^G c_j^\ell}{i!j!}\mathbb{E}_{x\sim\mathcal{N}(0,\mathbb{1}_d)}[h_i(w \cdot x)h_j(w \cdot v)]\Big|_{w=w_0} \\
=& \frac{\partial}{\partial w}\sum_{k=0}^{\infty}\frac{c_k^G c_k^\ell}{k!}\alpha_0^k\Big|_{w=w_0} = v_{\perp w_0}\left(\sum_{k=1}^{\infty}\frac{c_k^G c_k^\ell}{(k-1)!}\alpha_0^{k-1}\right) = v_{\perp w_0}\left(\sum_{k=k^*}^{\infty}\frac{c_k^G c_k^\ell}{(k-1)!}\alpha_0^{k-1}\right)
\end{aligned}
$$

where $v_{\perp w_0} := v - \alpha_0 w_0$ is the projection of the spike in the space orthogonal to $w_0$. Moreover, recall that $k^*$ is the information exponent of the population loss (2), in the sense of Definition 9. In the case of the spiked model defined in Section 2.2, we already know that $k^* = 4$. We keep the terms up to the sixth-order of expansion. Hence,

$$
\mathbb{E}\left[(\star)_1\right] = v_{\perp w_0}\left(\frac{c_4^G c_4^\ell}{3!}\alpha_0^3 + \frac{c_6^G c_6^\ell}{5!}\alpha_0^5\right) + o(\alpha_0^6).
$$

On the other hand, also have to take into account the component of the gradient in the direction of the initialization. For this second part, we get

$$
\begin{aligned}
\mathbb{E}[(\star)_2] =& w_0\,\mathbb{E}\left[\left.\frac{d}{d\lambda}L(\lambda w_0, x)\right|_{\lambda=1}\right] = w_0\,\mathbb{E}\left[G'(w_0 \cdot x)(w_0 \cdot x)\right] \\
=& w_0\,\mathbb{E}\left[G'(w_0 \cdot x)h_1(w_0 \cdot x)\ell(v \cdot x)\right] \\
=& w_0\sum_{i,j=0}^{\infty}\frac{c_i^{G'} c_j^\ell}{i!j!}\mathbb{E}_{x\sim\mathcal{N}(0,\mathbb{1}_d)}\left[h_i(w_0 \cdot x)h_1(w_0 \cdot x)h_j(v \cdot x)\right].
\end{aligned}
$$

By using Corollary 14 we can expand up to six order, obtaining

$$
\mathbb{E}[(\star)_2] = w_0\left(c_2^G c_0^\ell + \frac{c_4^G c_0^\ell}{3!}\alpha_0^4 + \frac{c_6^G c_4^\ell}{4!}\alpha_0^4 + \frac{c_6^G c_6^\ell}{5!}\alpha_0^6 + \frac{c_8^G c_6^\ell}{6!}\alpha_0^6\right) + o(\alpha_0^6).
$$

In conclusion, summing up the two contributions we simply have

$$\mathbb{E}[f] = v_{\perp w_0} \left( \frac{c_4^G c_4^\ell}{3!} \alpha_0^3 + \frac{c_6^G c_6^\ell}{5!} \alpha_0^5 \right) + w_0 \left( c_2^G c_0^\ell + \frac{c_4^G c_0^\ell}{3!} \alpha_0^4 + \frac{c_6^G c_4^\ell}{4!} \alpha_0^4 + \frac{c_6^G c_6^\ell}{5!} \alpha_0^6 + \frac{c_8^G c_6^\ell}{6!} \alpha_0^6 \right) + o(\alpha_0^6).$$

Therefore, it remains to compute $\mathbb{E}[g]$. To do so, it is sufficient to expand $G''$ in series of Hermite polynomials to obtain

$$\mathbb{E}[g] = w_0 \sum_{k=0}^\infty \frac{c_k^{G''} c_k^\ell}{k!} \alpha_0^k = w_0 \left( c_2^G c_0^\ell + \frac{c_6^G c_4^\ell}{4!} \alpha_0^4 + c_2^G c_0^\ell + \frac{c_8^G c_6^\ell}{6!} \alpha_0^6 \right) + o(\alpha_0^6).$$

Now that we have expanded both $\mathbb{E}[f]$ and $\mathbb{E}[g]$ up to sixth order, after a long but simple calculation, we can conclude by writing

$$\pi_1^2 = \|\widetilde{w}_1\|^2 = \frac{(c_4^G c_4^\ell)^2}{3!^2} \alpha_0^6 + o(\alpha_0^6) \quad \Longrightarrow \quad \alpha_1^2 = 1 - o(1).$$

This computation has shown that if the first step could possibly be infinitely large, then FastICA would perfectly learn the spike up to contributions which vanish when $d \to \infty$.

We observe now that happens when the iteration of FastICA consists only in the gradient term. The different update would be given by $\widetilde{w}_1 = \mathbb{E}[x\, G'(w_0 \cdot x)]$. In this case, it is not needed to expand up to sixth-order, since we can easily see that $\alpha_1^2$ scales as bad as the random initialisation. In particular,

$$\pi_1^2 = (c_2^G c_0^\ell)^2 \alpha_0^2 + o(\alpha_0^2), \; \|\widetilde{w}_1\|^2 = (c_2^G c_0^\ell)^2 + o(1) \quad \Longrightarrow \quad \alpha_1^2 = \alpha_0^2 + o(\alpha_0^2) = O\left(\frac{1}{d}\right).$$

$\square$

We are now going to prove Theorem 2, which makes explicit the sample complexity required to learn the spike in a single step with a large, although finite, amount of data points. Recall that, given $n$ samples, the FastICA iteration reads

$$\widetilde{w}_t = \frac{1}{n} \sum_{\nu=1}^n x^\nu\, G'(w_{t-1} \cdot x^\nu) - \frac{1}{n} \sum_{\nu=1}^n G''(w_{t-1} \cdot x^\nu) w_{t-1}.$$

*Proof of Theorem 2 (Finite batch-size).* Define $f := \frac{1}{n} \sum_{\nu=1}^n x^\nu\, G'(w_0 \cdot x^\nu)$ and $g := \frac{1}{n} \sum_{\nu=1}^n x^\nu\, G''(w_0 \cdot x^\nu) w_0$. The key ideas of this strategy are borrowed from the proof of Theorem 1 and 2 in Dandi et al. (2024), where they consider a teacher-student setup and Gaussian inputs. First of all, we compute the expectations of $\pi_1 = \widetilde{w}_1 \cdot v$ and $\|\widetilde{w}_1\|^2$. After that, we look at how much data we need to obtain concentration for both the quantities, namely what is the scaling for $n$ such that $\pi_1$ and $\|\widetilde{w}_1\|^2$ are approximated, up to a sufficiently small error, by their expectations. Then, we get the scaling for $\alpha_1^2$ by simply computing the ratio between $\mathbb{E}[\pi_1]^2$ and $\mathbb{E}[\|\widetilde{w}_1\|^2]$.

From now on, we consider $n = \Theta(d^{k^*-\delta})$, for $\delta \in [0, 2]$. In order to compute the expectation of $\pi_1$, we can use the calculation performed in the proof of Proposition 1 to obtain

$$\mathbb{E}[\pi_1] = v \cdot \mathbb{E}[f - g] = v \cdot \frac{1}{n} \sum_{\nu=1}^n \left( \mathbb{E}[x^\nu G'(w_0 \cdot x^\nu)] - \mathbb{E}[G''(w_0 \cdot x^\nu) w_0] \right)$$

$$= \frac{c_{k^*}^G c_{k^*}^\ell}{(k^* - 1)!} \alpha_0^{k^*-1} + o(\alpha_0^{k^*-1}),$$

where the last inequality is obtained by following the steps of Proposition 1 without explicitly using that $k^* = 4$. Indeed,

$$v \cdot \mathbb{E}[f] = c_0^\ell c_2^G \alpha_0 + \frac{c_{k^*}^G c_{k^*}^\ell}{(k^* - 1)!} \alpha_0^{k^*-1} + o(\alpha_0^{k^*-1}) \quad \text{and} \quad v \cdot \mathbb{E}[g] = c_0^\ell c_2^G \alpha_0 + o(\alpha_0^{k^*-1}).$$

We now have to compute $\mathbb{E}[\|w_1\|^2] = \mathbb{E}[\|f\|^2] + \mathbb{E}[\|g\|^2] - 2\,\mathbb{E}[f \cdot g]$. Since

$$\mathbb{E}[\|g\|^2] = \frac{1}{n^2}\mathbb{E}\left[\sum_{\nu,\nu'=1}^{n}(x^\nu \cdot x^{\nu'})\,G'(w_0 \cdot x^\nu)\,G'(w_0 \cdot x^{\nu'})\right]$$

$$= \frac{1}{n^2}\mathbb{E}\left[\sum_{\nu \neq \nu'}^{n}(x^\nu \cdot x^{\nu'})\,G'(w_0 \cdot x^\nu)\,G'(w_0 \cdot x^{\nu'})\right] + \frac{1}{n^2}\mathbb{E}\left[\sum_{\nu=1}^{n}\|x^\nu\|^2\,G'(w_0 \cdot x^\nu)^2\right]$$

$$= \frac{n(n-1)}{n^2}\|\mathbb{E}[g]\|^2 + \frac{1}{n}\mathbb{E}\left[\|x\|^2\,G'(w_0 \cdot x)^2\right]$$

and it is possible to do the same calculation for $\mathbb{E}[\|f\|^2]$ and $\mathbb{E}[f \cdot g]$, we get that

$$\mathbb{E}[\|\widetilde{w}_1\|^2] = \frac{n(n-1)}{n^2}\left(\|\mathbb{E}[g]\|^2 + \|\mathbb{E}[f]\|^2 - 2\,\mathbb{E}[g] \cdot \mathbb{E}[f]\right) \tag{14}$$
$$+ \frac{1}{n}\left(\mathbb{E}[\|x\|^2 G'(w_0 \cdot x)^2] + \mathbb{E}[G''(w_0 \cdot x)] - 2\,\mathbb{E}[(x \cdot w_0)G'(w_0 \cdot x)G''(w_0 \cdot x)]\right).$$

We can compute the scaling for the last three terms by using Lemma 12. On the other hand, the scaling for each of the first three terms is obtained by expanding $\mathbb{E}[f]$ and $\mathbb{E}[g]$ up to sixth-order and repeating the same calculation of Proposition 1 while avoiding the evaluation $k^* = 4$. We get that, for any $n$, we have

$$\mathbb{E}[\|w_1\|^2] = \frac{n(n-1)}{n^2}\left(\frac{(c_{k^*}^G c_{k^*}^\ell)^2}{(k^*-1)!^2}\alpha_0^{2(k^*-1)}\right)$$
$$+ \frac{1}{n}\left[d\left(\sum_{k \in 2\mathbb{N}+1}\frac{c_k^{G'} c_k^\ell}{k!}\right) + O(d\alpha_0^{k^*}) + \left(\sum_{k \in 2\mathbb{N}}\frac{c_k^{G''} c_k^\ell}{k!}\right) + O(\alpha_0^{k^*}) - 2c_0^{G''}c_1^{G'}c_0^\ell + O(1)\right]$$
$$= \frac{(c_{k^*}^G c_{k^*}^\ell)^2}{(k^*-1)!^2}\alpha_0^{2(k^*-1)} - \frac{1}{n^2}\left(\frac{(c_{k^*}^G c_{k^*}^\ell)^2}{(k^*-1)!^2}\alpha_0^{2(k^*-1)}\right) + \frac{1}{n}\left[d\left(\sum_{k \in 2\mathbb{N}+1}\frac{c_k^{G'} c_k^\ell}{k!}\right) + O(d\alpha_0^{k^*})\right.$$
$$+ \left.\left(\sum_{k \in 2\mathbb{N}}\frac{c_k^{G''} c_k^\ell}{k!}\right) + O(\alpha_0^4) - 2c_0^{G''}c_1^{G'}c_0^\ell + O(1)\right].$$

Hence, we start to keep into account the different choices of $\delta \in [0, 2]$. Recall that $n = \Theta(d^{k^*-\delta})$. We need to know that is the addendum which dominates in the previous formula, depending on $k^*$. By looking at the scaling of each addendum, one can see that there are two possible regimes, which correspond to the two cases $\alpha_0^{2(k^*-1)} \lessgtr d/n$. In particular, when $\delta = 0$, the signal coming from the contribution of the first line in (14) dominates. More precisely, we get that

$$\begin{cases} \delta \in (0, 2] \quad \Rightarrow \quad \mathbb{E}[\|w_1\|^2] = \Theta\left(\frac{d}{n}\right), \\ \delta = 0 \quad \Rightarrow \quad \mathbb{E}[\|w_1\|^2] = \frac{(c_{k^*}^G c_{k^*}^\ell)^2}{(k^*-1)!^2}\alpha_0^{2(k^*-1)} + o(\alpha_0^{2(k^*-1)}). \end{cases}$$

Hence, we have the scaling of the expectations of $\pi_1$ and $\|w_1\|^2$. It is left to see how much data samples are required to approximate well these expectations with high probability, i.e. for which sample complexity $\pi_1$ and $\|w_1\|^2$ concentrate. Thanks to Assumptions 1, 2 and 3, the proof is analogous to the one in Section B.4 from (Dandi et al., 2024), leading to:

$$|\pi_1 - \mathbb{E}[\pi_1]| \leq |f \cdot v - \mathbb{E}[f \cdot v]| + |g \cdot v - \mathbb{E}[g \cdot v]| = O\left(\frac{\log(n)}{\sqrt{n}}\right).$$

and

$$|\|w_1\|^2 - \mathbb{E}[\|w_1\|^2]| \leq |\|f\|^2 - \mathbb{E}[\|f\|^2]| + |\|g\|^2 - \mathbb{E}[\|g\|^2]| + 2|f \cdot g - \mathbb{E}[f \cdot g]|$$
$$= O\left(\frac{d\log(n)^6}{n\sqrt{n}} + \frac{\log(d)^6}{n\sqrt{d}} + \frac{\log(n)^2}{n} + \frac{\log(n)^{k^*}}{d^{(k^*-1)/2}\sqrt{n}}\right).$$

**Negative results:** We start by analysing what happens when $\delta \in (0, 2]$. We have with high probability that

$$\|w_1\|^2 \geq |\mathbb{E}[\|w_1\|^2]| - |\mathbb{E}[\|w_1\|^2] - \|w_1\|^2| = \Theta\left(\frac{d}{n}\right) - O\left(\frac{d\log(n)^6}{n\sqrt{n}} + \frac{\log(d)^6}{n\sqrt{d}} + \frac{\log(n)^2}{n} + \frac{\log(n)^{k^*}}{d^{(k^*-1)/2}\sqrt{n}}\right)$$

which implies that

$$\|w_1\|^2 = \Omega\left(\frac{d}{n}\right).$$

Moreover, with high probability,

$$|\pi_1| \leq |\mathbb{E}[\pi_1]| + |\mathbb{E}[\pi_1] - \pi_1| = O\left(\frac{1}{d^{(k^*-1)/2}}\right) + O\left(\frac{\log(n)}{\sqrt{n}}\right) \quad \Rightarrow \quad \pi_1 = O\left(\frac{1}{d^{(k^*-1)/2}}\right) + O\left(\frac{\log(n)}{\sqrt{n}}\right).$$

We have to split the reasoning in two cases.

**Case 1** : If $\delta \in (0, 1]$, with high probability $\pi_1 = O\left(\frac{1}{d^{(k^*-1)/2}}\right)$ and then

$$\alpha_1^2 = \frac{\pi_1^2}{\|w_1\|^2} = \frac{O\left(\frac{1}{d^{k^*-1}}\right)}{\Omega\left(\frac{d}{n}\right)} = \frac{O\left(\frac{1}{d^{k^*-1}}\right)}{\Omega\left(\frac{d^\delta}{d^{k^*-1}}\right)} = O\left(\frac{1}{d^\delta}\right).$$

It means that the overlap at step one is bounded by above by a quantity which vanishes when the dimension of the inputs $d$ goes to infinity, which is definitely a bad news for learning in high dimensions with FastICA. Note however that this case bridges smoothly the cases $\delta = 0$ and the case $\delta = 1$, which correspond to $d^{k^*-1}$, i.e. the regime in which online SGD is able to learn.

**Case 2:** If $\delta = [1, 2]$, the reason why FastICA does not learn is not an adverse ration between two random, although concentrated, quantities. In this case, the number of samples is not enough to reach concentration for $\pi_1$, that is $\pi_1 = O\left(\frac{\log(n)}{\sqrt{n}}\right)$. Therfore,

$$\alpha_1^2 = \frac{\pi_1^2}{\|w_1\|^2} = \frac{O\left(\frac{\text{polylog}(n)}{n}\right)}{\Omega\left(\frac{d}{n}\right)} = O\left(\frac{\text{polylog}(d)}{d}\right).$$

The latter negative result is *qualitatively* different from the previous one, since it is due to the lack of concentration of $\pi_1$. The spike is not recovered at all, meaning that the scaling of $\alpha_1$ is as bad as the one provided by the random weights initialisation.

**Positive result:** Let's consider $\delta = 0$, which means that we employee an extensive number of data points, namely $n = \Theta(d^{k^*})$. Then,

$$\|w_1\|^2 \leq |\mathbb{E}[\|w_1\|^2]| + |\mathbb{E}[\|w_1\|^2] - \|w_1\|^2| = \frac{(c_{k^*}^G c_{k^*}^\ell)^2}{(k^*-1)!^2}\alpha_0^{2(k^*-1)} + o(\alpha_0^{2(k*-1)}).$$

On the other hand, we also have the following lower bound:

$$|\pi_1| \geq |\mathbb{E}[\pi_1]| - |\mathbb{E}[\pi_1] - \pi_1| = \left|\frac{c_{k^*}^G c_{k^*}^\ell}{(k^*-1)!}\alpha_0^{k^*-1}\right| - o(\alpha_0^{k^*-1}).$$

In conclusion, we get

$$\alpha_1^2 = \frac{\pi_1^2}{\|w_1\|^2} \geq \frac{\frac{(c_{k^*}^G c_{k^*}^\ell)^2}{(k^*-1)!^2}\alpha_0^{2(k^*-1)} - o(\alpha_0^{2(k^*-1)})}{\frac{(c_{k^*}^G c_{k^*}^\ell)^2}{(k^*-1)!^2}\alpha_0^{2(k^*-1)} + o(\alpha_0^{2(k^*-1)})} = 1 - o(1),$$

meaning that in this regime, when $d \to +\infty$, the relevant direction is perfectly recovered by the first iteration of FastICA. □

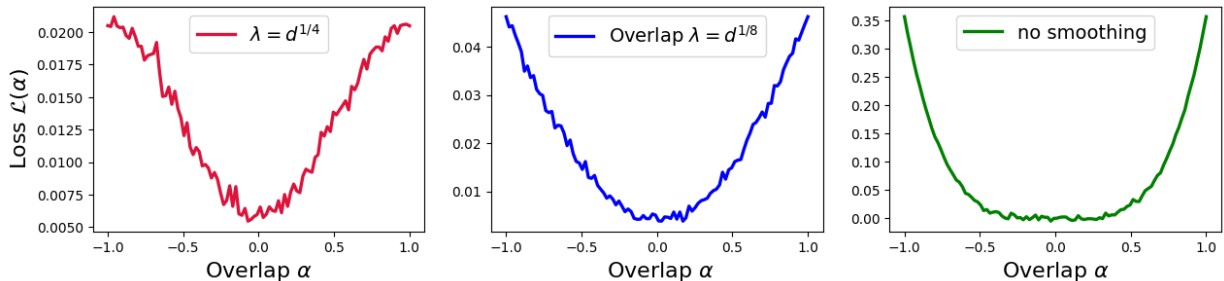

*Figure 6.* **Benefit of smoothing the loss depending on different values of the parameter** $\lambda$: We show the smoothed loss $\mathcal{L}_\lambda[G(w \cdot x)]$ for different scalings of the smoothing parameter $\lambda$. We can see the loss is most flat around the origin without smoothing, while increasing the smoothing parameter makes the loss more peaked around the origin, making it easier to escape the origin. See Appendix E for details.

## E. Smoothing the landscape

For implementation purposes, we start by writing the formula for the spherical gradient of the smoothed loss, that is

$$\nabla_{\text{sph}} L_\lambda(w, x) = \nabla_{\text{sph}} \mathcal{L}_\lambda(G(w \cdot x)) = x \, \mathbb{E}_{z_1} \left[ G' \left( \frac{w \cdot x + \lambda z_1 \|P_w^\perp x\|}{\sqrt{1 + \lambda^2}} \right) \left( \frac{1}{\sqrt{1 + \lambda^2}} - \frac{\lambda z_1 (w \cdot x)}{\|P_w^\perp x\| \sqrt{1 + \lambda^2}} \right) \right],$$

where $z_1$ is one of the components, for example the first one, of a random vector $z \sim \text{Unif}(\mathbb{S}^{d-2})$. This formula can be obtained by following the proof of Lemma 9 in (Damian et al., 2023). Note that we need to compute the spherical gradient in order to write the iteration of *smoothed online SGD*.

We now briefly comment on the benefit of smoothing the loss landscape, which is given by taking into account the expectation of the contrast function evaluated in the dot product between the (normalized) vector $w + \lambda z$ and the vector of the inputs $x$, instead of evaluating the loss in $s = w \cdot x$. The smoothing operator is defined in Definition 3. Recall that $z$ is a vector of white noise and note that in Theorem 4 the parameter $\lambda$ is large, namely it scales with the dimension $d$ of the inputs. The key intuition behind the smoothing operator is that, thanks to large $\lambda$, it allows to evaluate the loss function in regions that are far from the updated weights $w_t$, collecting non-local signal that alleviates the flatness of the saddle of the loss function at the beginning of learning, reducing the length of the search phase. In Figure 6 we plot the smoothed loss function, in the cases of $\lambda = d^{1/4}$, $\lambda = d^{1/8}$ and $\lambda = 0$. The first case corresponds to the optimal value of $\lambda$, that is, the one for which the optimal thresholds of Theorem 4 are obtained. The latter case corresponds to the absence of smoothing. It can be interesting to recall that for $\lambda \gg d^{1/4}$ we have the "noise' which dominates the "signal', in the sense of Equation (15), where the admissible values of $\lambda$ are chosen. For the plots in Figure 6 we have used $n = 10^4$ number of samples, $d = 10$ and $G(s) = -h_4(s)$.

In the following section we will go through the rigorous argument that formalizes the heuristic derivation presented in Section 3.5.1.

### E.1. Proof

We prove now Theorem 4, which provides the timescales for recovering the relevant direction with smoothed online SGD. Since the Gaussian version of this theorem has already been proved in (Damian et al., 2023), we will only emphasise the main differences between the Gaussian and non-Gaussian case, which are mainly due to the presence of the (bounded) likelihood ratio and the fact that the analysis cannot take into account only the information exponent $k_2^*$ of the contrast function, but also the information exponent of the population loss, and those in general are not equal. However, we will conclude that smoothed online SGD obtains the optimal sample complexity only in the matched case, which is for $k_1^* = k_2^*$.

*Proof of Theorem 4 (Escaping mediocrity).* The proof is made of two parts. The first part is dedicated to the computation of the scaling for the two main quantities that take part in the iteration of smoothed online SGD, i.e. the signal and the noise. The first part ends when we have collected all the ingredients to prove an analogue of the thesis of Lemma 15 in (Damian et al., 2023), which for us is given by (17). This formula guarantees that the overlap between the spike and the updated weight vector given by smoothed online SGD is precisely the sum of the previous overlap, the signal, the noise and a suitable martingale term. The second part is dedicated to the application of standard arguments in probability theory (for the case of vanilla and smoothed SGD see e.g. (Ben Arous et al., 2021) and (Damian et al., 2023) respectively), from which

it is straight forward to conclude that at some point, i.e. for a certain number of used data points, SGD reaches a regime in which the signal dominates the noise and the spike is recovered.

**Signal**: We start by studying the scaling of the population (spherical) gradient $\nabla_{\mathrm{sph}} L_\lambda(w)$. This scaling will depend on the information exponent $k_1^*$ of the population loss. To do so, consider the expansion as series of Hermite polynomials of the population loss

$$\mathcal{L}(w) = \sum_{k \geq k_1^*}^\infty \frac{c_k^G c_k^\ell}{k!} \alpha^k,$$

where $k_1^*$ is its information exponent. In Lemma 8 of (Damian et al., 2023) it is shown that we can apply the smoothing operator to each term and obtain that, for any $k \geq 0$,

$$\mathcal{L}_\lambda(\alpha^k) \approx s_k(\alpha, \lambda), \text{ with } s_k(\alpha, \lambda) := \frac{1}{(1+\lambda^2)^{k/2}} \begin{cases} \alpha^k & \alpha^2 \geq \frac{\lambda^2}{d}, \\ (\frac{\lambda^2}{d})^{k/2} & \alpha^2 \leq \frac{\lambda^2}{d} \text{ and } k \text{ is even}, \\ \alpha(\frac{\lambda^2}{d})^{(k-1)/2} & \alpha^2 \leq \frac{\lambda^2}{d} \text{ and } k \text{ is odd}. \end{cases}$$

By computation, we can obtain that the population (spherical) gradient is $\nabla_{\mathrm{sph}} L_\lambda(w) = (v - \alpha w)c_\lambda(\alpha)$, with

$$c_\lambda(\alpha) := \sum_{k \geq k_1^*} \frac{c_k^G c_k^\ell}{k!} \frac{d}{d\alpha} \mathcal{L}_\lambda(\alpha^k).$$

The scalar product of this population gradient term and the relevant direction, i.e. $v \cdot \nabla_{\mathrm{sph}} L_\lambda(w) = (1 - \alpha^2)c_\lambda(\alpha)$, is called *signal*. It can be proved, like in Lemma 10 of (Damian et al., 2023), that

$$c_\lambda(\alpha) \approx \frac{s_{k_1^*-1}(\alpha, \lambda)}{\sqrt{1+\lambda^2}}.$$

**Noise**: Now we take a look at the noise term. Fix $u \in \mathbb{S}^{d-1}$, with $u \perp w$. Recall that we are using the notation $g = \nabla_{\mathrm{sph}} L_\lambda(w, x)$. Since we have assumed that the likelihood ratio $\ell$ is bounded (e.g. as the case of inputs drawn according to the noisy ICA model (5)), we get

$$\mathbb{E}\left[(u \cdot g)^2\right] = \mathbb{E}_{x \sim \mathcal{N}(0, \mathbb{1}_d)}\left[(u \cdot g)^2 \ell(v \cdot x)\right] \leq \|\ell\|_\infty \mathbb{E}_{x \sim \mathcal{N}(0, \mathbb{1}_d)}\left[(u \cdot g)^2\right] \tag{15}$$

$$\lesssim \frac{\min\left(1 + \lambda^2, \sqrt{d}\right)^{-(k_2^*-1)}}{1 + \lambda^2} \lesssim (1 + \lambda^2)^{-k_2^*}, \tag{16}$$

where we have used the fact that $\lambda \in [1, d^{1/4}]$ and Corollary 1 in (Damian et al., 2023). In the last inequality, thanks to the choice $\lambda \in [1, d^{1/4}]$, we using the fact that

$$\min\left(1 + \lambda^2, \sqrt{d}\right)^{-(k_2^*-1)} = \left(1 + \lambda^2\right)^{-(k_2^*-1)}$$

Note that we have obtain an explicit dependence on the information exponent $k_2^*$ of the contrast function. We can estimate now the second moment of the smoothed spherical gradient, i.e. $\mathbb{E}[\|g\|^2]$, which we call *noise*. To do so, recall that $g$ is a spherical gradient and then of course $g \perp w$. To get an upper bound on the scaling of the noise, it is sufficient to fix an orthonormal basis in $\mathbb{R}^d$ such that $e_1 = w$. Then,

$$\mathbb{E}[\|g\|^2] = \sum_{i=2}^d \mathbb{E}\left[(e_i \cdot g)^2\right] \lesssim d\, O((1 + \lambda^2)^{-k_2^*}).$$

Now that we have the desired upper bound, it is possible to compute the scaling for the expectation of some $p$- moments of $\|g\|^2$, for $p$-sufficiently large. This corresponds to Corollary 2 in (Damian et al., 2023).

By considering both the scaling of the signal and the noise, we can now state the non-Gaussian version of Lemma 15 in (Damian et al., 2023), which tells us that if $\alpha_{t+1} := v \cdot w_{t+1}$ and $w_{t+1} := \frac{w_t + \eta g_t}{\|w_t + \eta g_t\|}$, we obtain that with high probability

$$\alpha_{t+1} = \alpha_t + \eta(1 - \alpha_t^2)c_\lambda(\alpha_t) + Z_t + O(\mathbb{E}[\|g_t\|^2]). \tag{17}$$

where $\{Z_t\}_{t\in\mathbb{N}}$ is a martingale term with zero mean defined as

$$Z_t := \eta\left(g_t \cdot v\right) - \mathbb{E}\left[\eta\left(g_t \cdot v\right)\right] + r_t - \mathbb{E}\left[r_t\right]$$

and

$$r_t := \left[\alpha_t + \eta\left(g_t \cdot v\right)\right]\left(\frac{1}{\sqrt{1 + \eta^2\|g_t\|^2}} - 1\right).$$

Hence, at each step of smoothed online SGD, the updated overlap between the new weight vector and the spike is nothing but the previous overlap plus a martingale term, the signal - whose scaling depends on $k_1^*$ - and the noise - whose scaling depends on $k_2^*$. We have reached the setting of Lemma 16 of (Damian et al., 2023). From here on, the proof can be concluded by applying the argument that translates the signal and noise estimated into sample complexities, introduced in the setting of vanilla online SGD by (Ben Arous et al., 2021). The proofs of Lemma 16 and Lemma 17 from (Damian et al., 2023) can be repeated straight forwardly, leading to the required sample complexities for recovering the spike. $\square$

## F. Additional numerics

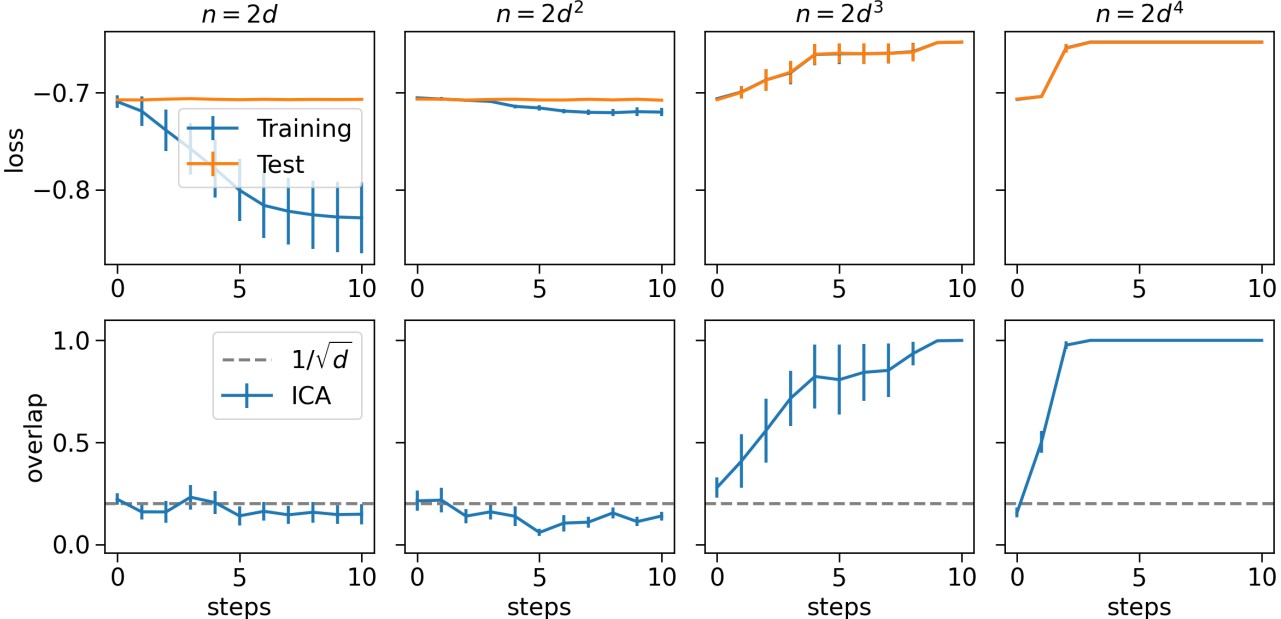

Figure 7. **Performance of FastICA on noisy ICA model with Rademacher prior on the latent *without* resampling the training data at each step**. *Parameters*: input dimension $d = 25$, signal-to-noise ratio $\beta = 5$, online learning.

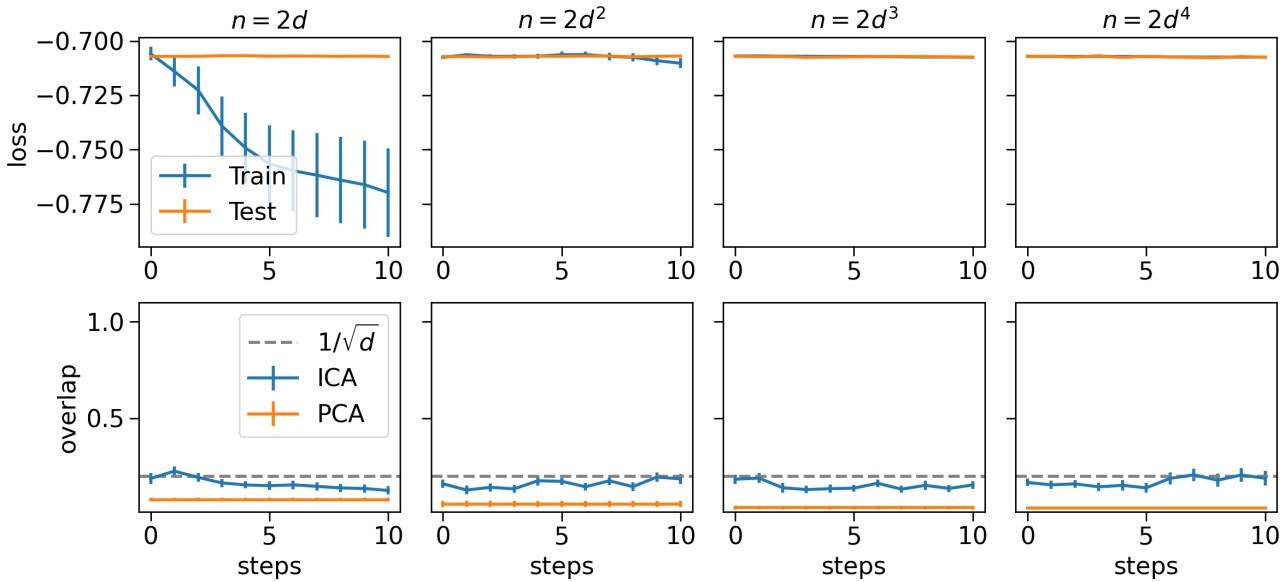

Figure 8. **Performance of online FastICA on noisy ICA model with the "3not4" prior on the latent variable**. Under this prior, the latent has mean zero, unit variance, a non-trivial third-order cumulant, but zero fourth-order cumulant; see Appendix A.5 for details. *Parameters*: input dimension $d = 25$, signal-to-noise ratio $\beta = 5$, online learning.

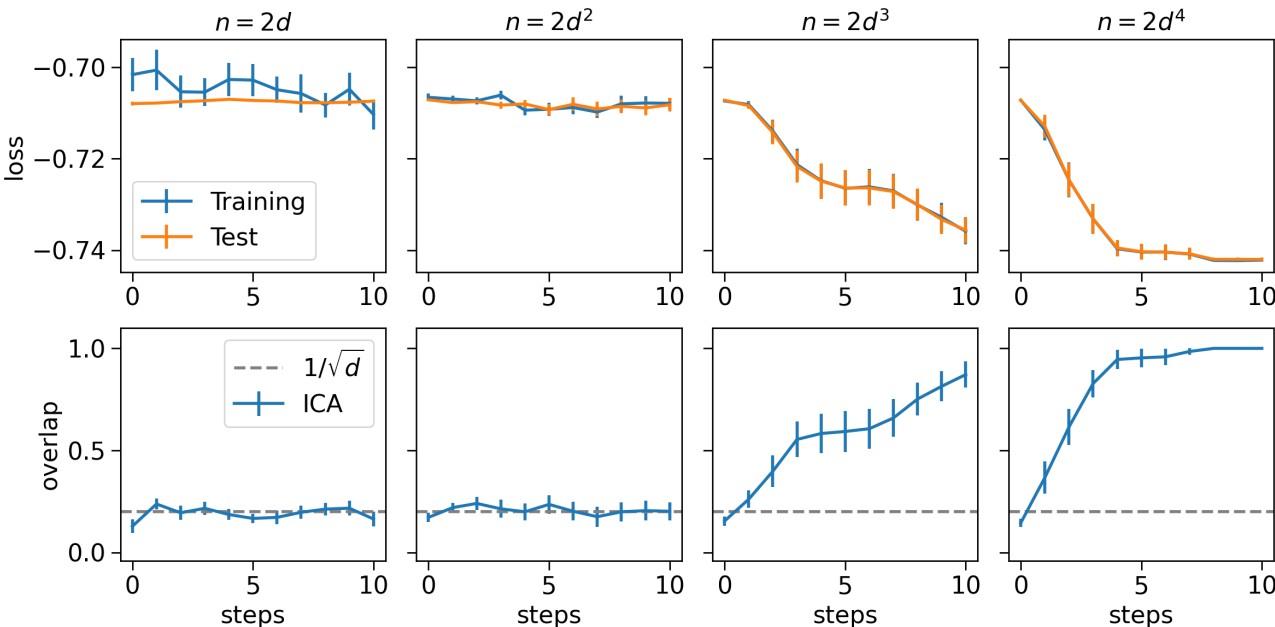

Figure 9. **Performance of online FastICA on noisy ICA model with Laplace prior on the latent**. *Parameters*: input dimension $d = 25$, signal-to-noise ratio $\beta = 5$, online learning.

