# OpenReview forum: "Feature learning from non-Gaussian inputs: the case of Independent Component Analysis in high dimensions"
_ICML.cc/2025/Conference — ICML 2025 spotlightposter_

### Official Review · Reviewer_eH2y · 2025-03-12

**Overall Recommendation:** 4

**Summary:**

This paper investigates the unsupervised learning method ICA as a simplified framework for feature learning in (deep) CNNs. In particular, the authors derive sample complexity thresholds for escaping the search phase of FastICA and SGD, considering a toy model of a dataset sampled from an isotropic distribution perturbed by a single non-Gaussian direction. Their results highlight the poor performance of ICA compared to SGD, with the latter being able, in principle, to achieve the computational threshold.

**Claims And Evidence:**

Every claim is supported by convincing evidence.

**Essential References Not Discussed:**

I am not aware of any relevant related work that has been omitted.

**Experimental Designs Or Analyses:**

I have reviewed the experimental details related to the figures in the paper. The explanations are clear, and I have no issues to discuss.

**Methods And Evaluation Criteria:**

Although the proposed analysis focuses on a relatively simple model to derive quantitative results, it provides valuable insights into the performance of two popular algorithms used in practical applications.

**Other Comments Or Suggestions:**

- Inconsistent notation in Fig. 1: $\mathbf{B}$ is used instead of $\mathbf{(b)}$
- I suggest writing explicitly what quantity is represented in the legends of Fig. 4

**Other Strengths And Weaknesses:**

The presentation is overall clear and easy to comprehend. The assumptions are correctly stated and well-motivated, and the results are supported by numerical simulations. One potential weakness of this work is the relative simplicity of the model, which perturbs the isotropic Gaussian distribution along a single direction and is distant from realistic image datasets. However, I do not consider this a major weakness, as it is compensated by the originality of the results in relation to the existing literature, providing a baseline for investigating more complex settings.

**Questions For Authors:**

I do not have additional questions

**Relation To Broader Scientific Literature:**

The paper extends the findings of Auddy & Yuan (2024) on the optimal algorithmic threshold for ICA, rigorously analyzing the performance of the most widely used algorithms in practice and comparing them to the known optimal result. Moreover, their proofs extend recent methods developed in the context of supervised learning with Gaussian inputs to the unsupervised non-Gaussian setting.

**Theoretical Claims:**

I checked the correctness of the proofs of Proposition 1, Theorem 2 and Theorem 4. I do not have any issues to discuss.

---

> ### Author Rebuttal · Authors · 2025-03-28
>
> Thank you for your careful feedback and for having checked both the numerical experiments and the proofs. Your suggestions will definitely help to clarify the paper.
>
> > One potential weakness of this work is the relative simplicity of the model. However, I do not consider this a major weakness, as it is compensated by the originality of the results in relation to the existing literature, providing a baseline for investigating more complex settings.
>
> Indeed, our data model is a simple single-index model. We decided to focus on single-index models in this paper because of the prevalence of **Gaussian** single index studies in the recent past (Ben Arous et al. JMLR '21, Bietti et al. NeurIPS '22, Damian et al. NeurIPS '23, Damian et al. arXiv:2403.05529, Bardone & Goldt ICML '24, Wang et al. NeurIPS '24). Here, we provide an extension to the non-Gaussian case, which is closer to realistic data. We are considering on an extension to multi-spike models for our future work.
>
> > Inconsistent notation in Fig. 1.B is used instead of (b).
>
> Thanks for noticing, we fixed this.
>
> > I suggest writing explicitly what quantity is represented in the legends of Fig.4.
>
> Thank you, we have added a title to the legends to stress that these are the widths of the patches extracted from ImageNet images.

---

> > ### Comment · Reviewer_eH2y · 2025-04-09
> >
> > Thank you for your reply and further clarification.

---

### Official Review · Reviewer_Jmtu · 2025-03-12

**Overall Recommendation:** 4

**Summary:**

The study quantifies the sample complexity of two learning algorithms: FastICA, Stochastic Gradient Descent (SGD). The key results are the following: FastICA requires at least $n \gtrsim d^4$ samples to recover a single non-Gaussian direction in high-dimensional inputs. SGD outperforms FastICA in feature learning, achieving better sample complexity, particularly when smoothing techniques are used ($n \gtrsim d^2$). In the real-world dataset ImageNet, the strong non-Gaussianity of image data helps mitigate FastICA’s inefficiency.

## update after rebuttal
Thank you for the authors' response. The authors have addressed my main concerns. After reviewing the other reviewers' comments, I will keep my score and recommend accepting this paper.

**Claims And Evidence:**

Yes, for the most part, the claims are well-supported by theoretical derivations and empirical experiments. However, more empirical justification to explain that "the growth of the excess kurtosis might compensate the poor sample complexity of FastICA in practice" would be useful, currently it is not fully explored.

**Essential References Not Discussed:**

Not founded.

**Experimental Designs Or Analyses:**

The experimental setups are well-documented, specifying parameter choices, dataset details, and batch sizes.
Different regimes for FastICA (Figure 2) and SGD (Figure 3) are clearly explained. The spiked cumulant model used for evaluation is well-motivated. One potential issue is the color scales of the filters in Figure 1(b) could be adjusted to better match those in Figure 1(a) for better visual comparability.

**Methods And Evaluation Criteria:**

Yes, the methods and evaluation criteria are appropriate. The ImageNet experiment is a reasonable real-world test, although the authors could have explored other high-dimensional datasets for broader generalizability.

**Other Comments Or Suggestions:**

1. It would be beneficial to provide additional explanations on why Independent Component Analysis learns similar filters as deep convolutional neural networks in Fig. 1. Specifically, more details are needed to illustrate how this similarity can be observed directly in the figure.
2. Titles should begin with capitalized initial letters.
3. "We finally **investigated** whether FastICA also exhibits an extended search phase on real data." -> "We finally **investigate** whether FastICA also exhibits an extended search phase on real data." to maintain present tense consistency.
4. I think perhaps highlighting the generality of non-Gaussian distributions (e.g., using Cramér's decomposition theorem) would emphasise the importance of this work more significantly.

**Other Strengths And Weaknesses:**

Strength:
1. The paper is rigorously written with strong theoretical contributions, which advances the understanding of feature learning dynamics in high-dimensional settings.
2. The link between ICA and CNN feature learning is well-motivated and provides a fresh perspective on the emergence of structured filters.
3. The theoretical results are logically structured, featuring clear mathematical formulations and providing essential technical background information.

Weaknesses:
1. An explicit emphasis in the title that this is a paper analysing sample complexity might make it clearer to the reader.
2. The writing of the introduction can be improved to increase readability.  For example, the first paragraph of the introduction gives a somewhat disjointed impression. After discussing Gabor filters, it shifts to a discussion of non-Gaussianity. SGD is also introduced quite abruptly. The transitions between these topics are non-smooth, making it difficult for the reader to follow the logical flow. A similar issue arises in the section describing the contributions of the paper, where several key terms and expressions appear without prior introduction. As a result, it is challenging for the reader to grasp the main focus of the work upon first reading the introduction. Smoother transitions and clearer structuring would enhance readability and coherence.
3. The explanation of why SGD benefits from smoothing the contrast function could be clearer, especially for readers less familiar with statistical-to-computational gaps. Additional visualisations would be better.
4. The analysis is focused on learning one non-Gaussian feature. A discussion on how multiple features interact would be valuable.

**Questions For Authors:**

1. The batch sizes are described as $n=d^2$, $d^3+\delta$, $d^4$, but it is unclear how $\delta=0.2$ was chosen. Is there any reason to choose $\delta=0.2$? Does it generalize to different values?
2. The Hermite expansion assumes $f(x)$ is square-integrable. However, many ICA contrast functions are not bounded (e.g., excess kurtosis). Could the authors clarify whether the expansion holds for all practical contrast functions?
3. In Figure 3, the learning rate $\eta$ is defined based on $k_1^*$ and $k_2^*$, but how to tune it in practice?

**Relation To Broader Scientific Literature:**

The findings in this paper could be relevant for designing more efficient feature extraction methods in deep learning.

**Theoretical Claims:**

The proofs appear mathematically rigorous. I have some related questions, see “Questions For Authors”.

---

> ### Author Rebuttal · Authors · 2025-03-28
>
> Thank you for your detailed feedback and your useful comments, which spurred us to run an additional experiment to link ICA with deep CNNs, and to provide two additional plots to provide intuition on the effect of smoothing, and on the intermediate regime of FastICA. We start with these points before addressing your remaining comments below. We hope our reply alleviates any remaining concern; if not, please let us know, otherwise we would really appreciate it if you increased your score. Thank you for your time!
>
> > Readability of the introduction & explanations on why Independent Component Analysis learns similar filters as deep CNNs in Fig. 1.
>
> Thank you for this suggestion -- we now realise the introduction jumped from deep CNNs to ICA too quickly. To make the connection more clearly, we have conducted an **additional experiment**: we trained three deep CNNs (AlexNet, Resnet18 and DenseNet121) on ImageNet and computed the excess kurtosis of the dot products $s$ between first-layer convolutional filters and ImageNet patches. We found that the excess kurtosis of $s$, which corresponds to the objective function of ICA, sharply increases after about 1000 steps of SGD, precisely when Gabor filters form in the first layer. This demonstrates empirically that neural networks seek out non-Gaussian directions when Gabor filters form, akin to what happens when running ICA. We summarise our results in a revised Figure 1 in https://figshare.com/s/0e72e797306c1a3f216a, and that we hope will clarify the motivation for studying ICA.
>
> > The explanation of why SGD benefits from smoothing the contrast function could be clearer [...]
>
> The key intuition behind the smoothing operator is that, thanks to large $\lambda$ (than scales with the input dimension), it allows to evaluate the loss function in regions that are far from the iterate weight $w_t$, collecting non-local signal that alleviates the flatness of the saddle around $\alpha=0$ of $\mathcal L$, reducing the length of the search phase.  A new plot, Fig 3 of https://figshare.com/s/0e72e797306c1a3f216a, shows the reduced flatness of the smoothed loss at the origin.
>
> > Choice of $\delta$ in the batch sizes $n=d^2, d^{3 + \delta}$, and $d^4$
>
> We could have chosen any $\delta \in (0,1)$, since Theorem 2 implies that the 'intermediate' regime shown in Fig. 2 (middle) holds for a batch size of $d^2 \ll n \ll d^3$. We will add Fig 2 of https://figshare.com/s/0e72e797306c1a3f216a which shows the behaviour for  $\delta = 0.5, 0.8$.
>
> > The Hermite expansion, square integrability and contrast functions.
>
> We merely need the contrast functions to be square-integrable with respect to the standard normal distribution $P_0$, so we need that the growth of $G$ is less than exponential. In general, we don't require they are integrable with respect to the Lebesgue measure on the real line (e.g., excess kurtosis is not Lebesgue integrable).
>
> > In Figure 3, the learning rate $\eta$ is defined based on $k^*_1$, $k^*_2$, but how to tune it in practice?
>
> Our theorem 4 offers a way to make a reasonable guess for $\eta$ as follows: $k^*_2$ is the information exponent of the contrast function and is therefore known. Meanwhile, $k^*_1$ depends on the likelihood and hence on the data distribution. However, we have the bound $k^*_1 \geq k^*_2$. Furthermore, if we assume that data has a non-trivial fourth-order cumulant (a mild assumption in practice), we can set $k^*_1=4$ and hence our theorem gives a recommendation on how to scale the learning rate in practice.
>
> > The analysis is focused on learning one non-Gaussian feature. A discussion on how multiple features interact would be valuable.
>
> We decided to focus on single-index models in this paper because of the prevalence of Gaussian single index studies in the recent past (Ben Arous et al. (2021), Damian et al. (2023), Damian et al. (2024), Bardone & Goldt (2024)). We are considering on an extension to multi-spike models for our future work.
>
> > Ensuring present tense consistency
>
> Thanks, we fixed this.
>
> > Highlighting the generality of non-Gaussian distributions (Cramér's decomposition theorem) [...]
>
> Yes, we agree that Cramér's theorem emphasizes the importance of studying non-Gaussianities, we will discuss it.
>
> > Additional empirical justification to explain that "the growth of the excess kurtosis might compensate the poor sample complexity of FastICA in practice"
>
> We will move the plot from the appendix to Fig. 4 with ImageNet results. We agree that it does not fully explain the behaviour of FastICA on ImageNet, but the growth of excess kurtosis with input dimension hints at important finite-size effects, which are out of the purview of our asymptotic theory.
>
> > Explicit emphasis in the title that this paper analyses sample complexity [...].
>
> Thank you for this suggestion; we will consider clarifying the title should the paper be accepted. We have already made changes to clarify our objectives and motivations in the introduction (see below).

---

### Official Review · Reviewer_oLx5 · 2025-03-14

**Overall Recommendation:** 4

**Summary:**

Motivated by empirical observations that features learned by deep convolutional networks resemble those recovered by independent components analysis (ICA), this paper presents a concrete algorithmic sample complexity bound for various algorithms for recovering a non-Gaussian direction from $d$-dimensional data. Notably, the paper establishes a sample complexity bound of $n \gtrsim d^4$ for a popular ICA algorithm (FastICA), and demonstrates that SGD in fact outperforms this algorithm on this task, attaining the information-theoretic lower bound of $n \approx d^2$ (albeit through a smoothed loss).

**Claims And Evidence:**

The theoretical claims in this paper are very well-supported. Various lower bounds and fundamental limits from prior work are outlined. The upper bounds in the paper are accompanied by simple numerical experiments demonstrating that they capture the correct scaling.

**Essential References Not Discussed:**

Not that I'm aware of.

**Experimental Designs Or Analyses:**

A sufficient description of experiment set-ups is contained in the main paper, with additional details in Appendix A.

**Methods And Evaluation Criteria:**

The numerical experiments are helpful for contextualizing the theory, and are documented in the appendix.

**Other Comments Or Suggestions:**

- The authors should remember to put in the Impact Statement.

- This paper shows that ICA is computationally and statistically tractable. Something that might be helpful is a discussion of why identifying non-Gaussian directions might be desirable or might happen automatically, since it seems the ultimate goal is to demonstrate that neural networks might implicitly learn non-Gaussian directions. A concrete mathematical statement here might be to show recovering these directions improves generalization error, analogous to how weak recovery improves generalization in standard Gaussian single-index models. This is likely a hard problem, but food for thought.

**Other Strengths And Weaknesses:**

I think this paper is written well and quite straightforward to digest, despite the technicality of the theoretical tools. The main insights and arguments are likely of interest to the feature learning theory community. I have described some of the strengths that stood out to me earlier. I don't see any glaring weaknesses to the paper, beyond the possible restrictiveness of the single non-Gaussian direction model--however, this is rather trite, as algorithmic results for other multi-index settings are yet generally ill-understood.

**Questions For Authors:**

None outstanding.

**Relation To Broader Scientific Literature:**

This work belongs to the general category of literature concerned with understanding the statistical properties of machine learning problems and algorithms from the lens of "feature learning", i.e. how algorithms actually recover predictive features that, e.g., go beyond the linear regime (captured by the information exponent). This work in particular has potential value to the community in two main ways: 1. as stated in the paper, empirical observations have shown particular structure learned by CNNs that seem to match those given by earlier ICA algorithms--the results in this paper make this connection explicit, 2. the sample complexity bounds in this paper seem to properly close the statistical-to-computational gap for the non-Gaussian direction recovery problem, showing that both FastICA and vanilla online SGD are suboptimal, and that smoothed-loss SGD closes the gap.

**Theoretical Claims:**

I did not check the proofs entirely. However, I checked the proof strategies of the main Theorems and they make sense to me. The numerical experiments also are helpful to sanity check the correctness of the theorems (such as demonstrating the necessity of smoothing the loss for SGD).

---

> ### Author Rebuttal · Authors · 2025-03-28
>
> Thank you for your accurate comments and for the attention to the supplementary material, including the strategies of the proofs. Your suggestions offered valuable food for thought.
>
> >  I don't see any glaring weaknesses to the paper, beyond the possible restrictiveness of the single non-Gaussian direction model -- however, this is rather trite, as algorithmic results for other multi-index settings are yet generally ill-understood.
>
> Thanks for your observation. We decided to focus on single-index models in this paper because of the prevalence of Gaussian single index studies in the recent past (Ben Arous et al. JMLR '21, Bietti et al. NeurIPS '22, Damian et al. NeurIPS '23, Damian et al. arXiv:2403.05529, Bardone & Goldt ICML '24, Wang et al. NeurIPS '24). We are considering on an extension to multi-spike models for our future work.
>
> > The authors should remember to put in the Impact Statement.
>
> Thank you for the reminder, we will add it.
>
> > Something that might be helpful is a discussion of why identifying non-Gaussian directions might be desirable or might happen automatically. A concrete mathematical statement here might be to show recovering these directions improves generalization error.
>
> The reviewer raises a really interesting point, which is why it is desirable to learn non-Gaussian directions in the first place. Fascinating work in theoretical neuroscience has established that natural images have a highly non-Gaussian statistical structure: while pixel intensities themselves may follow roughly Gaussian distributions, the relationships between pixels are strongly non-Gaussian. Specifically, natural images are sparse in certain bases: edges, contours, and textures are more prevalent than random pixel noise. Gabor-like filters, i.e. non-Gaussian directions, are efficient at capturing these features that yield sparse image representations, see for example reviews such as Simoncelli & Olshausen, Annu. Rev. Neurosci. 24:1193–216 (2001) or the book by Hyvärinen, Hurri and Hoyer (2009).
>
> More recently, there have been a series of works investigating the importance of non-Gaussian input structures in machine learning from a mathematical perspective, showing that neural networks will learn them if they are relevant for the task, i.e. if they improve generalisation; see in particular Ingrosso & Goldt, PNAS '22; Bardone & Goldt, ICML '24; Lufkin et al. NeurIPS '24.
>
> We will add an extended discussion of this issue to the revised version of the paper, should it be accepted.

---

### Decision · Program_Chairs · 2025-05-01

**Decision:**

Accept (spotlight poster)

**Comment:**

This work is motivated by the link between independent component analysis (ICA) and CNN feature learning, and proposes to advance the understanding of feature learning through the simple model of ICA, which seeks to recover the most non-Gaussian projection. It compares FastICA, a standard ICA algorithm, and stochastic gradient descent (SGD), which is commonly used in the training of deep neural networks, in terms of sample complexity. The derived performance bounds imply that, at the computational threshold of quadratic sample complexity, FastICA does not have concentrated update to allow it escape effectively from random initialization, whereas SGD, when properly smoothed in a data-dependent way, is able to recover the signal.

As remarked by the reviewers, the study is presented in a carefully structured manuscript, containing meaningful theoretical results under well-motivated assumptions, supported by clearly explained experiments. The simplicity of the theoretical setup is compensated by the originality of the results. The reviewers have maintained their original positive ratings after the rebuttal, during which the authors have provided convincing replies to the reviewers’ comments.